# A Unified Bellman Optimality Principle Combining Reward Maximization and Empowerment

**Felix Leibfried,   Sergio Pascual-Díaz,   Jordi Grau-Moya**
PROWLER.io
Cambridge, UK
{felix,sergio.diaz,jordi}@prowler.io

## Abstract

Empowerment is an information-theoretic method that can be used to intrinsically motivate learning agents. It attempts to maximize an agent's control over the environment by encouraging visiting states with a large number of reachable next states. Empowered learning has been shown to lead to complex behaviors, without requiring an explicit reward signal. In this paper, we investigate the use of empowerment in the presence of an extrinsic reward signal. We hypothesize that empowerment can guide reinforcement learning (RL) agents to find good early behavioral solutions by encouraging highly empowered states. We propose a unified Bellman optimality principle for empowered reward maximization. Our empowered reward maximization approach generalizes both Bellman's optimality principle as well as recent information-theoretical extensions to it. We prove uniqueness of the empowered values and show convergence to the optimal solution. We then apply this idea to develop off-policy actor-critic RL algorithms which we validate in high-dimensional continuous robotics domains (MuJoCo). Our methods demonstrate improved initial and competitive final performance compared to model-free state-of-the-art techniques.

## 1   Introduction

In reinforcement learning [62] (RL), agents identify policies to collect as much reward as possible in a given environment. Recently, leveraging parametric function approximators has led to tremendous success in applying RL to high-dimensional domains such as Atari games [40] or robotics [56]. In such domains, inspired by the policy gradient theorem [63, 13], actor-critic approaches [36, 41] attain state-of-the-art results by learning both a parametric policy and a value function.

Empowerment is an information-theoretic framework where agents maximize the mutual information between an action sequence and the state that is obtained after executing this action sequence from some given initial state [26, 27, 53]. It turns out that the mutual information is highest for such initial states where the number of reachable next states is largest. Policies that aim for high empowerment can lead to complex behavior, e.g. balancing a pole in the absence of any explicit reward signal [23].

Despite progress on learning empowerment values with function approximators [42, 12, 49], there has been little attempt in the combination with reward maximization, let alone in utilizing empowerment for RL in the high-dimensional domains it has become applicable just recently. We therefore propose a unified principle for reward maximization and empowerment, and demonstrate that empowered signals can boost RL in large-scale domains such as robotics. In short, our contributions are:

- a generalized Bellman optimality principle for joint reward maximization and empowerment,
- a proof for unique values and convergence to the optimal solution for our novel principle,
- empowered actor-critic methods boosting RL in MuJoCo compared to model-free baselines.

## 2 Background

### 2.1 Reinforcement Learning

In the discrete RL setting, an agent, being in state $s \in \mathcal{S}$, executes an action $a \in \mathcal{A}$ according to a behavioral policy $\pi_{\text{behave}}(a|s)$ that is a conditional probability distribution $\pi_{\text{behave}} : \mathcal{S} \times \mathcal{A} \to [0,1]$. The environment, in response, transitions to a successor state $s' \in \mathcal{S}$ according to a (probabilistic) state-transition function $\mathcal{P}(s'|s, a)$, where $\mathcal{P} : \mathcal{S} \times \mathcal{A} \times \mathcal{S} \to [0,1]$. Furthermore, the environment generates a reward signal $r = \mathcal{R}(s, a)$ according to a reward function $\mathcal{R} : \mathcal{S} \times \mathcal{A} \to \mathbb{R}$. The agent's aim is to maximize its expected future cumulative reward with respect to the behavioral policy $\max_{\pi_{\text{behave}}} \mathbb{E}_{\pi_{\text{behave}}, \mathcal{P}} \left[ \sum_{t=0}^{\infty} \gamma^t r_t \right]$, with $t$ being a time index and $\gamma \in (0,1)$ a discount factor. Optimal expected future cumulative reward values for a given state $s$ obey then the following recursion:

$$V^\star(s) = \max_a \left( \mathcal{R}(s, a) + \gamma \mathbb{E}_{\mathcal{P}(s'|s,a)} \left[ V^\star(s') \right] \right) =: \max_a Q^\star(s, a), \qquad (1)$$

referred to as Bellman's optimality principle [4], where $V^\star$ and $Q^\star$ are the optimal value functions.

### 2.2 Empowerment

Empowerment is an information-theoretic method where an agent executes a sequence of $k$ actions $\vec{a} \in \mathcal{A}^k$ when in state $s \in \mathcal{S}$ according to a policy $\pi_{\text{empower}}(\vec{a}|s)$ which is a conditional probability distribution $\pi_{\text{empower}} : \mathcal{S} \times \mathcal{A}^k \to [0,1]$. This is slightly more general than in the RL setting where only a single action is taken upon observing a certain state. The agent's aim is to identify an optimal policy $\pi_{\text{empower}}$ that maximizes the mutual information $I\left[\vec{A}, S' \middle| s\right]$ between the action sequence $\vec{a}$ and the state $s'$ to which the environment transitions after executing $\vec{a}$ in $s$, formulated as:

$$E^\star(s) = \max_{\pi_{\text{empower}}} I\left[\vec{A}, S' \middle| s\right] = \max_{\pi_{\text{empower}}} \mathbb{E}_{\pi_{\text{empower}}(\vec{a}|s)\mathcal{P}^{(k)}(s'|s,\vec{a})} \left[ \log \frac{p(\vec{a}|s', s)}{\pi_{\text{empower}}(\vec{a}|s)} \right]. \qquad (2)$$

Here, $E^\star(s)$ refers to the optimal empowerment value and $\mathcal{P}^{(k)}(s'|s, \vec{a})$ to the probability of transitioning to $s'$ after executing the sequence $\vec{a}$ in state $s$, where $\mathcal{P}^{(k)} : \mathcal{S} \times \mathcal{A}^k \times \mathcal{S} \to [0,1]$. Importantly, $p(\vec{a}|s', s) = \frac{\mathcal{P}^{(k)}(s'|s,\vec{a})\pi_{\text{empower}}(\vec{a}|s)}{\sum_{\vec{a}} \mathcal{P}^{(k)}(s'|s,\vec{a})\pi_{\text{empower}}(\vec{a}|s)}$ is the inverse dynamics model of $\pi_{\text{empower}}$. The implicit dependency of $p$ on the optimization argument $\pi_{\text{empower}}$ renders the problem non-trivial.

From an information-theoretic perspective, optimizing for empowerment is equivalent to maximizing the capacity [58] of an information channel $\mathcal{P}^{(k)}(s'|s, \vec{a})$ with input $\vec{a}$ and output $s'$ w.r.t. the input distribution $\pi_{\text{empower}}(\vec{a}|s)$, as outlined in the following [11, 10]. Define the functional $I_f(\pi_{\text{empower}}, \mathcal{P}^{(k)}, q) := \mathbb{E}_{\pi_{\text{empower}}(\vec{a}|s)\mathcal{P}^{(k)}(s'|s,\vec{a})} \left[ \log \frac{q(\vec{a}|s',s)}{\pi_{\text{empower}}(\vec{a}|s)} \right]$, where $q$ is a conditional probability $q : \mathcal{S} \times \mathcal{S} \times \mathcal{A}^k \to [0,1]$. Then the mutual information is recovered as a special case of $I_f$ with $I\left[\vec{A}, S' \middle| s\right] = \max_q I_f(\pi_{\text{empower}}, \mathcal{P}^{(k)}, q)$ for a given $\pi_{\text{empower}}$. The maximum argument

$$q^\star(\vec{a}|s', s) = \frac{\mathcal{P}^{(k)}(s'|s, \vec{a})\pi_{\text{empower}}(\vec{a}|s)}{\sum_{\vec{a}} \mathcal{P}^{(k)}(s'|s, \vec{a})\pi_{\text{empower}}(\vec{a}|s)} \qquad (3)$$

is the true Bayesian posterior $p(\vec{a}|s', s)$—see [10] Lemma 10.8.1 for details. Similarly, maximizing $I_f(\pi_{\text{empower}}, \mathcal{P}^{(k)}, q)$ with respect to $\pi_{\text{empower}}$ for a given $q$ leads to:

$$\pi_{\text{empower}}^\star(\vec{a}|s) = \frac{\exp\left(\mathbb{E}_{\mathcal{P}^{(k)}(s'|s,\vec{a})} \left[\log q(\vec{a}|s', s)\right]\right)}{\sum_{\vec{a}} \exp\left(\mathbb{E}_{\mathcal{P}^{(k)}(s'|s,\vec{a})} \left[\log q(\vec{a}|s', s)\right]\right)}. \qquad (4)$$

As explained e.g. in [10] page 335 similar to [46]. The above yields the subsequent proposition.

**Proposition 1** *Maximum Channel Capacity. Iterating through Equations (3) and (4) by computing $q$ given $\pi_{\text{empower}}$ and vice versa in an alternating fashion converges to an optimal pair $(q^\star, \pi_{\text{empower}}^\star)$ that maximizes the mutual information $\max_{\pi_{\text{empower}}} I\left[\vec{A}, S' \middle| s\right] = I_f(\pi_{\text{empower}}^\star, \mathcal{P}^{(k)}, q^\star)$. The convergence rate is $\mathcal{O}(1/N)$, where $N$ is the number of iterations, for any initial $\pi_{\text{empower}}^{\text{ini}}$ with support in $\mathcal{A}^k \ \forall s$— see [10] Chapter 10.8 and [11, 16]. This is known as Blahut-Arimoto algorithm [2, 7].*

**Remark.** *Empowerment is similar to curiosity concepts of predictive information that focus on the mutual information between the current and the subsequent state [6, 48, 69, 61, 43, 54].*

## 3 Motivation: Combining Reward Maximization with Empowerment

The Blahut-Arimoto algorithm presented in the previous section solves empowerment for low-dimensional discrete settings but does not readily scale to high-dimensional or continuous state-action spaces. While there has been progress on learning empowerment values with parametric function approximators [42], how to combine it with reward maximization or RL remains open. In principle, there are two possibilities for utilizing empowerment. The first is to directly use the policy $\pi^\star_{\text{empower}}$ obtained in the course of learning empowerment values $E^\star(\boldsymbol{s})$. The second is to train a behavioral policy to take an action in each state such that the expected empowerment value of the next state is highest (requiring $E^\star$-values as a prerequisite). Note that the two possibilities are conceptually different. The latter seeks states with a large number of reachable next states [23]. The first, on the other hand, aims for high mutual information between actions and the subsequent state, which is not necessarily the same as seeking highly empowered states [42].

We hypothesize empowered signals to be beneficial for RL, especially in high-dimensional environments and at the beginning of the training process when the initial policy is poor. In this work, we therefore combine reward maximization with empowerment inspired by the two behavioral possibilities outlined in the previous paragraph. Hence, we focus on the cumulative RL setting rather than the non-cumulative setting that is typical for empowerment. We furthermore use one-step empowerment as a reference, i.e. $k = 1$, because cumulative one-step empowerment learning leads to high values in such states where the number of possibly reachable next states is high, and preserves hence the original empowerment intuition *without* requiring a multi-step policy—see Section 4.3. The first idea is to train a policy that trades off reward maximization and *learning* cumulative empowerment:

$$\max_{\pi_{\text{behave}}} \mathbb{E}_{\pi_{\text{behave}},\mathcal{P}} \left[ \sum_{t=0}^{\infty} \gamma^t \left( \alpha \mathcal{R}(\boldsymbol{s}_t, \boldsymbol{a}_t) + \beta \log \frac{p(\boldsymbol{a}_t | \boldsymbol{s}_{t+1}, \boldsymbol{s}_t)}{\pi_{\text{behave}}(\boldsymbol{a}_t | \boldsymbol{s}_t)} \right) \right], \tag{5}$$

where $\alpha \geq 0$ and $\beta \geq 0$ are scaling factors, and $p$ indicates the inverse dynamics model of $\pi_{\text{behave}}$ in line with Equation (3). Note that $p$ depends on the optimization argument $\pi_{\text{behave}}$, similar to ordinary empowerment, leading to a non-trivial Markov decision problem (MDP).

The second idea is to learn cumulative empowerment values *a priori* by solving Equation (5) with $\alpha = 0$ and $\beta = 1$. The outcome of this is a policy $\pi^\star_{\text{empower}}$ (and its inverse dynamics model $p$) that can be used to construct an intrinsic reward signal which is then added to the external reward:

$$\max_{\pi_{\text{behave}}} \mathbb{E}_{\pi_{\text{behave}},\mathcal{P}} \left[ \sum_{t=0}^{\infty} \gamma^t \left( \alpha \mathcal{R}(\boldsymbol{s}_t, \boldsymbol{a}_t) + \beta \mathbb{E}_{\pi^\star_{\text{empower}}(\boldsymbol{a}|\boldsymbol{s}_t)\mathcal{P}(\boldsymbol{s}'|\boldsymbol{s}_t,\boldsymbol{a})} \left[ \log \frac{p(\boldsymbol{a}|\boldsymbol{s}', \boldsymbol{s}_t)}{\pi^\star_{\text{empower}}(\boldsymbol{a}|\boldsymbol{s}_t)} \right] \right) \right]. \tag{6}$$

Importantly, Equation (6) poses an ordinary MDP since the reward signal is merely extended by another stationary state-dependent signal.

Both proposed ideas require to solve the novel MDP as specified in Equation (5). In Section 4, we therefore prove the existence of unique values and convergence of the corresponding value iteration scheme (including a grid world example). We also show how our formulation generalizes existing formulations from the literature. In Section 5, we carry our ideas over to high-dimensional continuous state-action spaces by devising off-policy actor-critic-style algorithms inspired by the proposed MDP formulation. We evaluate our novel actor-critic-style algorithms in MuJoCo demonstrating better initial and competitive final performance compared to model-free state-of-the-art baselines.

## 4 Joint Reward Maximization and Empowerment Learning in MDPs

We state our main theoretical result *in advance*, proven in the remainder of this section (an intuition follows): the solution to the MDP from Equation (5) implies unique optimal values $V^\star$ obeying the Bellman recursion

$$\begin{aligned}
V^\star(\boldsymbol{s}) &= \max_{\pi_{\text{behave}}} \mathbb{E}_{\pi_{\text{behave}},\mathcal{P}} \left[ \sum_{t=0}^{\infty} \gamma^t \left( \alpha \mathcal{R}(\boldsymbol{s}_t, \boldsymbol{a}_t) + \beta \log \frac{p(\boldsymbol{a}_t | \boldsymbol{s}_{t+1}, \boldsymbol{s}_t)}{\pi_{\text{behave}}(\boldsymbol{a}_t | \boldsymbol{s}_t)} \right) \Big| \boldsymbol{s}_0 = \boldsymbol{s} \right] \\
&= \max_{\pi_{\text{behave}},q} \mathbb{E}_{\pi_{\text{behave}}(\boldsymbol{a}|\boldsymbol{s})} \left[ \alpha \mathcal{R}(\boldsymbol{s}, \boldsymbol{a}) + \mathbb{E}_{\mathcal{P}(\boldsymbol{s}'|\boldsymbol{s},\boldsymbol{a})} \left[ \beta \log \frac{q(\boldsymbol{a}|\boldsymbol{s}', \boldsymbol{s})}{\pi_{\text{behave}}(\boldsymbol{a}|\boldsymbol{s})} + \gamma V^\star(\boldsymbol{s}') \right] \right] \\
&= \beta \log \sum_{\boldsymbol{a}} \exp \left( \frac{\alpha}{\beta} \mathcal{R}(\boldsymbol{s}, \boldsymbol{a}) + \mathbb{E}_{\mathcal{P}(\boldsymbol{s}'|\boldsymbol{s},\boldsymbol{a})} \left[ \log q^\star(\boldsymbol{a}|\boldsymbol{s}', \boldsymbol{s}) + \frac{\gamma}{\beta} V^\star(\boldsymbol{s}') \right] \right),
\end{aligned} \tag{7}$$

where

$$q^\star(\boldsymbol{a}|\boldsymbol{s}',\boldsymbol{s}) = \frac{\mathcal{P}(\boldsymbol{s}'|\boldsymbol{s},\boldsymbol{a})\pi^\star_{\text{behave}}(\boldsymbol{a}|\boldsymbol{s})}{\sum_{\boldsymbol{a}} \mathcal{P}(\boldsymbol{s}'|\boldsymbol{s},\boldsymbol{a})\pi^\star_{\text{behave}}(\boldsymbol{a}|\boldsymbol{s})} = p(\boldsymbol{a}|\boldsymbol{s}',\boldsymbol{s}) \qquad (8)$$

is the inverse dynamics model of the optimal behavioral policy $\pi^\star_{\text{behave}}$ that assumes the form:

$$\pi^\star_{\text{behave}}(\boldsymbol{a}|\boldsymbol{s}) = \frac{\exp\left(\frac{\alpha}{\beta}\mathcal{R}(\boldsymbol{s},\boldsymbol{a}) + \mathbb{E}_{\mathcal{P}(\boldsymbol{s}'|\boldsymbol{s},\boldsymbol{a})}\left[\log q^\star(\boldsymbol{a}|\boldsymbol{s}',\boldsymbol{s}) + \frac{\gamma}{\beta}V^\star(\boldsymbol{s}')\right]\right)}{\sum_{\boldsymbol{a}}\exp\left(\frac{\alpha}{\beta}\mathcal{R}(\boldsymbol{s},\boldsymbol{a}) + \mathbb{E}_{\mathcal{P}(\boldsymbol{s}'|\boldsymbol{s},\boldsymbol{a})}\left[\log q^\star(\boldsymbol{a}|\boldsymbol{s}',\boldsymbol{s}) + \frac{\gamma}{\beta}V^\star(\boldsymbol{s}')\right]\right)}, \qquad (9)$$

where the denominator is just $\exp((1/\beta)V^\star(\boldsymbol{s}))$. While the remainder of this section explains how Equations (7) to (9) are derived in detail, it can be insightful to understand at a high level what makes our formulation non-trivial. The difficulty is that the inverse dynamics model $p = q^\star$ depends on the optimal policy $\pi^\star_{\text{behavioral}}$ and vice versa leading to a non-standard optimal value identification problem. Proving the existence of $V^\star$-values and how to compute them poses therefore our main theoretical contribution, and implies the existence of at least one $(q^\star, \pi^\star_{\text{behave}})$-pair that satisfies the recursive relationship of Equations (8) and (9). This proof is given in Section 4.1 and leads naturally to a value iteration scheme to compute optimal values in practice. The convergence of this scheme is proven in Section 4.2 and we also demonstrate value learning in a grid world example—see Section 4.3. In Section 4.4, we elucidate how our formulation generalizes and relates to existing MDP formulations.

## 4.1 Existence of Unique Optimal Values

Following the second line from Equation (7), let's define the Bellman operator $B_\star : \mathbb{R}^{|\mathcal{S}|} \to \mathbb{R}^{|\mathcal{S}|}$ as

$$B_\star V(\boldsymbol{s}) := \max_{\pi_{\text{behave}},q} \mathbb{E}_{\pi_{\text{behave}}(\boldsymbol{a}|\boldsymbol{s})}\left[\alpha\mathcal{R}(\boldsymbol{s},\boldsymbol{a}) + \mathbb{E}_{\mathcal{P}(\boldsymbol{s}'|\boldsymbol{s},\boldsymbol{a})}\left[\beta\log\frac{q(\boldsymbol{a}|\boldsymbol{s}',\boldsymbol{s})}{\pi_{\text{behave}}(\boldsymbol{a}|\boldsymbol{s})} + \gamma V(\boldsymbol{s}')\right]\right]. \qquad (10)$$

**Theorem 1** *Existence of Unique Optimal Values. Assuming a bounded reward function $\mathcal{R}$, the optimal value vector $V^\star$ as given in Equation (7) exists and is a unique fixed point $V^\star = B_\star V^\star$ of the Bellman operator $B_\star$ from Equation (10).*

**Proof.** The proof of Theorem 1 comprises three steps. First, we prove for a given $(q, \pi_{\text{behave}})$-pair the existence of unique values $V^{(q,\pi_{\text{behave}})}$ which obey the following recursion

$$V^{(q,\pi_{\text{behave}})}(\boldsymbol{s}) = \mathbb{E}_{\pi_{\text{behave}}(\boldsymbol{a}|\boldsymbol{s})}\left[\alpha\mathcal{R}(\boldsymbol{s},\boldsymbol{a}) + \mathbb{E}_{\mathcal{P}(\boldsymbol{s}'|\boldsymbol{s},\boldsymbol{a})}\left[\beta\log\frac{q(\boldsymbol{a}|\boldsymbol{s}',\boldsymbol{s})}{\pi_{\text{behave}}(\boldsymbol{a}|\boldsymbol{s})} + \gamma V^{(q,\pi_{\text{behave}})}(\boldsymbol{s}')\right]\right]. \qquad (11)$$

This result is obtained through Proposition 2 following [5, 51, 18] where we show that the value vector $V^{(q,\pi_{\text{behave}})}$ is a unique fixed point of the operator $B_{q,\pi_{\text{behave}}} : \mathbb{R}^{|\mathcal{S}|} \to \mathbb{R}^{|\mathcal{S}|}$ given by

$$B_{q,\pi_{\text{behave}}}V(\boldsymbol{s}) := \mathbb{E}_{\pi_{\text{behave}}(\boldsymbol{a}|\boldsymbol{s})}\left[\alpha\mathcal{R}(\boldsymbol{s},\boldsymbol{a}) + \mathbb{E}_{\mathcal{P}(\boldsymbol{s}'|\boldsymbol{s},\boldsymbol{a})}\left[\beta\log\frac{q(\boldsymbol{a}|\boldsymbol{s}',\boldsymbol{s})}{\pi_{\text{behave}}(\boldsymbol{a}|\boldsymbol{s})} + \gamma V(\boldsymbol{s}')\right]\right]. \qquad (12)$$

Second, we prove in Proposition 3 that solving the right hand side of Equation (10) for the pair $(q, \pi_{\text{behave}})$ can be achieved with a Blahut-Arimoto-style algorithm in line with [16]. Third, we complete the proof in Proposition 4 based on Proposition 2 and 3 by showing that $V^\star = \max_{\pi_{\text{behave}},q} V^{(q,\pi_{\text{behave}})}$, where the vector-valued max-operator is well-defined because both $\pi_{\text{behave}}$ and $q$ are conditioned on $\boldsymbol{s}$. The proof completion follows again [5, 51, 18]. $\qquad\square$

**Proposition 2** *Existence of Unique Values for a Given $(q, \pi_{\text{behave}})$-Pair. Assuming a bounded reward function $\mathcal{R}$, the value vector $V^{(q,\pi_{\text{behave}})}$ as given in Equation (11) exists and is a unique fixed point $V^{(q,\pi_{\text{behave}})} = B_{q,\pi_{\text{behave}}} V^{(q,\pi_{\text{behave}})}$ of the Bellman operator $B_{q,\pi_{\text{behave}}}$ from Equation (12).*

As opposed to the Bellman operator $B_\star$, the operator $B_{q,\pi_{\text{behave}}}$ does *not* include a max-operation that incurs a non-trivial recursive relationship between optimal arguments. The proof for existence of unique values follows hence standard methodology [5, 51, 18] and is given in Appendix A.1.

**Proposition 3** *Blahut-Arimoto for One Value Iteration Step. Assuming that $\mathcal{R}$ is bounded, the maximization problem $\max_{\pi_{\text{behave}},q}$ from Equation (10) in the Bellman operator $B_\star$ can be solved for $(q, \pi_{\text{behave}})$ by iterating through the following two equations in an alternating fashion:*

$$q^{(m)}(\boldsymbol{a}|\boldsymbol{s}',\boldsymbol{s}) = \frac{\mathcal{P}(\boldsymbol{s}'|\boldsymbol{s},\boldsymbol{a})\pi^{(m)}_{\text{behave}}(\boldsymbol{a}|\boldsymbol{s})}{\sum_{\boldsymbol{a}}\mathcal{P}(\boldsymbol{s}'|\boldsymbol{s},\boldsymbol{a})\pi^{(m)}_{\text{behave}}(\boldsymbol{a}|\boldsymbol{s})}, \qquad (13)$$

$$\pi_{\text{behave}}^{(m+1)}(\boldsymbol{a}|\boldsymbol{s}) = \frac{\exp\left(\frac{\alpha}{\beta}\mathcal{R}(\boldsymbol{s},\boldsymbol{a}) + \mathbb{E}_{\mathcal{P}(\boldsymbol{s}'|\boldsymbol{s},\boldsymbol{a})}\left[\log q^{(m)}(\boldsymbol{a}|\boldsymbol{s}',\boldsymbol{s}) + \frac{\gamma}{\beta}V(\boldsymbol{s}')\right]\right)}{\sum_{\boldsymbol{a}}\exp\left(\frac{\alpha}{\beta}\mathcal{R}(\boldsymbol{s},\boldsymbol{a}) + \mathbb{E}_{\mathcal{P}(\boldsymbol{s}'|\boldsymbol{s},\boldsymbol{a})}\left[\log q^{(m)}(\boldsymbol{a}|\boldsymbol{s}',\boldsymbol{s}) + \frac{\gamma}{\beta}V(\boldsymbol{s}')\right]\right)}, \tag{14}$$

*where $m$ is the iteration index. The convergence rate is $\mathcal{O}(1/M)$ for arbitrary initial $\pi_{\text{behave}}^{(0)}$ with support in $\mathcal{A}\ \forall \boldsymbol{s}$. $M$ is the total number of iterations. The complexity for a single $\boldsymbol{s}$ is $\mathcal{O}(M|\mathcal{S}||\mathcal{A}|)$.*

**Proof Outline.** The problem in Proposition 3 is mathematically similar to the maximum channel capacity problem [58] from Proposition 1 and proving convergence follows similar steps that we outline here—details can be found in Appendix A.2. First, we prove that optimizing the right-hand side of Equation (10) w.r.t. $q$ for a given $\pi_{\text{behave}}$ results in Equation (13) according to [10] Lemma 10.8.1. Second, we prove that optimizing w.r.t. $\pi_{\text{behave}}$ for a given $q$ results in Equation (14) following standard techniques from variational calculus and Lagrange multipliers. Third, we prove convergence to a global maximum when iterating alternately through Equations (13) and (14) following [16].

**Proposition 4** *Completing the Proof of Theorem 1. The optimal value vector is given by $V^\star = \max_{\pi_{\text{behave}},q} V^{(q,\pi_{\text{behave}})}$ and is a unique fixed point $V^\star = B_\star V^\star$ of the Bellman operator $B_\star$.*

Completing the proof of Theorem 1 requires two ingredients: the existence of unique $V^{(q,\pi_{\text{behave}})}$-values for any $(q,\pi_{\text{behave}})$-pair as proven in Proposition 2, and the fact that the optimal Bellman operator can be expressed as $B_\star = \max_{\pi_{\text{behave}},q} B_{q,\pi_{\text{behave}}}$ where $\max_{\pi_{\text{behave}},q}$ is the max-operator from Proposition 3. The proof follows then standard methodology [5, 51, 18], see Appendix A.3.

### 4.2 Value Iteration and Convergence to Optimal Values

In the previous section, we have proven the existence of unique optimal values $V^\star$ that are a fixed point of the Bellman operator $B_\star$. This section devises a value iteration scheme based on the operator $B_\star$ and proves its convergence. We commence by a corollary to express $B_\star$ more concisely.

**Corollary 1** *Optimal Bellman Operator. The operator $B_\star$ from Equation (10) can be written as*

$$B_\star V(\boldsymbol{s}) = \beta \log \sum_{\boldsymbol{a}} \exp\left(\frac{\alpha}{\beta}\mathcal{R}(\boldsymbol{s},\boldsymbol{a}) + \mathbb{E}_{\mathcal{P}(\boldsymbol{s}'|\boldsymbol{s},\boldsymbol{a})}\left[\log q^{\text{converged}}(\boldsymbol{a}|\boldsymbol{s}',\boldsymbol{s}) + \frac{\gamma}{\beta}V(\boldsymbol{s}')\right]\right), \tag{15}$$

*where $q^{\text{converged}}(\boldsymbol{a}|\boldsymbol{s}',\boldsymbol{s})$ is the result of the converged Blahut-Arimoto scheme from Proposition 3.*

This result is obtained by plugging the converged solution $\pi_{\text{behave}}^{\text{converged}}$ from Equation (14) into Equation (10) and leads naturally to a two-level value iteration algorithm that proceeds as follows: the outer loop updates the values $V$ by applying Equation (15) repeatedly; the inner loop applies the Blahut-Arimoto algorithm from Proposition 3 to identify $q^{\text{converged}}$ required for the outer value update.

**Theorem 2** *Convergence to Optimal Values. Assuming bounded $\mathcal{R}$ and let $\epsilon \in \mathbb{R}$ be a positive number such that $\epsilon < \frac{\eta}{1-\gamma}$ where $\eta = \alpha \max_{\boldsymbol{s},\boldsymbol{a}} |\mathcal{R}(\boldsymbol{s},\boldsymbol{a})| + \beta \log |\mathcal{A}|$. If the value iteration scheme with initial values of $V(\boldsymbol{s}) = 0\ \forall \boldsymbol{s}$ is run for $i \geq \left\lceil \log_\gamma \frac{\epsilon(1-\gamma)}{\eta} \right\rceil$ iterations, then $\left\| V^\star - B_\star^{(i)} V \right\|_\infty \leq \epsilon$, where the notation $B_\star^{(i)} V$ means to apply $B_\star$ to $V$ $i$-times consecutively.*

**Proof.** Via a sequence of inequalities, one can show that the following holds true: $\left\| V^\star - B_\star^{(i)} V \right\|_\infty \leq \gamma \left\| V^\star - B_\star^{(i-1)} V \right\|_\infty \leq \gamma^i \left\| V^\star - V \right\|_\infty \leq \gamma^i \frac{1}{1-\gamma}\eta$—see Appendix A.4 for a more detailed derivation. This implies that if $\epsilon \geq \gamma^i \frac{1}{1-\gamma}\eta$ then $i \geq \left\lceil \log_\gamma \frac{\epsilon(1-\gamma)}{\eta} \right\rceil$ presupposing $\epsilon < \frac{\eta}{1-\gamma}$. $\square$

**Conclusion.** *Together, Theorems 1 and 2 prove that our proposed value iteration scheme convergences to optimal values $V^\star$ in combination with a corresponding optimal pair $(q^\star, \pi_{\text{behave}}^\star)$ as described at the beginning of this section in the third line of Equation (7) and in Equations (8) and (9) respectively. The overal complexity is $\mathcal{O}(iM|\mathcal{S}|^2|\mathcal{A}|)$ where $i$ and $M$ refer to outer and inner iterations.*

**Remark.** *Our value iteration is required for both objectives from Section 3 to combine reward maximization with empowerment. Equation (5) motivated our scheme in the first place, whereas Equation (6) requires cumulative empowerment values without reward maximization ($\alpha = 0$, $\beta = 1$).*

### 4.3   Practical Verification in a Grid World Example

In order to practically verify our value iteration scheme from the previous section, we conduct experiments on a grid world example. The outcome is shown in Figure 1 demonstrating how different configurations for $\alpha$ and $\beta$, that steer cumulative reward maximization versus empowerment learning, affect optimal values $V^\star$. Importantly, the experiments show that our proposal to learn cumulative one-step empowerment values recovers the original intuition of empowerment in the sense that high values are assigned to states where many other states can be reached and low values to states where the number of reachable next states is low, *but without* the necessity to maintain a multi-step policy.

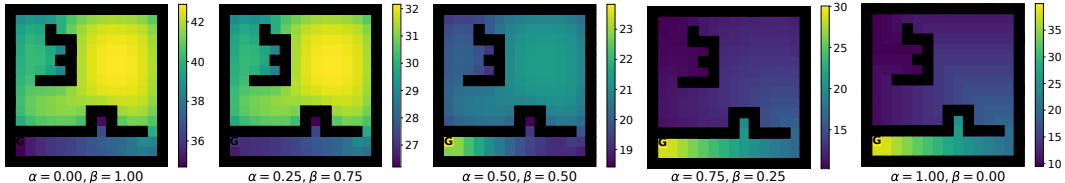

Figure 1: Value Iteration for a Grid World Example. The agent aims to arrive at the goal 'G' in the lower left—detailed information regarding the setup can be found in Appendix C.1. The plots show optimal values for different $\alpha$ and $\beta$: $\alpha$ increases from left to right while $\beta$ decreases. The leftmost values show raw cumulative empowerment learning ($\alpha = 0.0, \beta = 1.0$). High values are assigned to states where many other states can be reached, i.e. the upper right; and low values to states where the number of reachable next states is low, i.e. close to corners and dead ends. The rightmost values recover ordinary cumulative reward maximization ($\alpha = 1.0, \beta = 0.0$) assigning high values to states close to the goal and low values to states far away from the goal.

### 4.4   Generalization of and Relation to Existing MDP formulations

Our Bellman operator $B_\star$ from Equation (10) relates to prior work as follows (see also Appendix A.5).

- Ordinary value iteration [52] is recovered as a special case for $\alpha = 1$ and $\beta = 0$.
- Cumulative one-step empowerment is recovered as a special case for $\alpha = 0$ and $\beta = 1$, with *non-cumulative* one-step empowerment [29] as a further special case of the latter ($\gamma \to 0$).
- When setting $q(\boldsymbol{a}|\boldsymbol{s}', \boldsymbol{s}) = q(\boldsymbol{a}|\boldsymbol{s})$, using a distribution that is *not* conditioned on $\boldsymbol{s}'$ and *omitting* maximizing w.r.t. $q$, one recovers as a special case the soft Bellman operator presented e.g. in [51]. Note that this soft Bellman operator also occurred in numerous other work on MDP formulations and RL [3, 14, 45, 55, 33].
- As a special case of the previous, when $q(\boldsymbol{a}|\boldsymbol{s}', \boldsymbol{s}) = \mathcal{U}(\mathcal{A})$ is the uniform distribution in action space, one recovers cumulative entropy regularization [70, 44, 34] that inspired algorithms such as soft Q-learning [20] and soft actor-critic [21, 22].
- When dropping the conditioning on $\boldsymbol{s}'$ and $\boldsymbol{s}$ by setting $q(\boldsymbol{a}|\boldsymbol{s}', \boldsymbol{s}) = q(\boldsymbol{a})$ but *without omitting* maximization w.r.t. $q$, one recovers a formulation similar to [65] based on mutual-information regularization [59, 60, 17, 31] that spurred RL algorithms such as [30, 19, 32].
- When replacing $q(\boldsymbol{a}|\boldsymbol{s}', \boldsymbol{s})$ with $q(\boldsymbol{a}|\boldsymbol{s}_\prime, \boldsymbol{a}_\prime)$, where $\boldsymbol{s}_\prime$ and $\boldsymbol{a}_\prime$ refers to the state-action pair of the previous time step, one recovers a formulation similar to [64] based on the information-theoretic principle of directed information [38, 28, 39].

## 5   Scaling to High-Dimensional Environments

In the previous section, we presented a novel Bellman operator in combination with a value iteration scheme to combine reward maximization and empowerment. In this section, by leveraging parametric function approximators, we validate our ideas in high-dimensional state-action spaces and when there is no prior knowledge of the state-transition function. In Section 5.1, we devise novel actor-critic algorithms for RL based on our MDP formulation since they are naturally capable of handling both continuous state and action spaces. In Section 5.2, we practically confirm that empowerment can boost RL in the high-dimensional robotics simulator domain of MuJoCo using deep neural networks.

## 5.1 Empowered Off-Policy Actor-Critic Methods with Parametric Function Approximators

Contemporary off-policy actor-critic approaches for RL [36, 1, 15] follow the policy gradient theorem [63, 13] and learn two parametric function approximators: one for the behavioral policy $\pi_\phi(\boldsymbol{a}|\boldsymbol{s})$ with parameters $\phi$, and one for the state-action value function $Q_\theta(\boldsymbol{s}, \boldsymbol{a})$ of the parametric policy $\pi_\phi$ with parameters $\theta$. The policy learning objective usually assumes the form: $\max_\phi \mathbb{E}_{\boldsymbol{s} \sim \mathcal{D}} \left[ \mathbb{E}_{\pi_\phi(\boldsymbol{a}|\boldsymbol{s})} \left[ Q_\theta(\boldsymbol{s}, \boldsymbol{a}) \right] \right]$, where $\mathcal{D}$ refers to a replay buffer [37] that stores collected state transitions from the environment. Following [21], Q-values are learned most efficiently by introducing another function approximator $V_\psi$ for state values of $\pi_\phi$ with parameters $\psi$ using the objective:

$$\min_\theta \mathbb{E}_{\boldsymbol{s}, \boldsymbol{a}, r, \boldsymbol{s}' \sim \mathcal{D}} \left[ \left( Q_\theta(\boldsymbol{s}, \boldsymbol{a}) - (\alpha r + \gamma V_\psi(\boldsymbol{s}')) \right)^2 \right], \qquad (16)$$

where $(\boldsymbol{s}, \boldsymbol{a}, r, \boldsymbol{s}')$ refers to an environment interaction sampled from the replay buffer ($r$ stands for the observed reward signal). We multiply $r$ by the scaling factor $\alpha$ from our formulation because Equation (16) can be directly used for the parametric methods we propose. Learning policy parameters $\phi$ and value parameters $\psi$ requires however novel objectives with two additional approximators: one for the inverse dynamics model $p_\chi(\boldsymbol{a}|\boldsymbol{s}', \boldsymbol{s})$ of $\pi_\phi$, and one for the transition function $\mathcal{P}_\xi(\boldsymbol{s}'|\boldsymbol{s}, \boldsymbol{a})$ (with parameters $\chi$ and $\xi$ respectively). While the necessity for $p_\chi$ is clear, e.g. from inspecting Equation (5), the necessity for $\mathcal{P}_\xi$ will fall into place shortly as we move forward.

In order to preserve a clear view, let's define the quantity $f(\boldsymbol{s}, \boldsymbol{a}) := \mathbb{E}_{\mathcal{P}_\xi(\boldsymbol{s}'|\boldsymbol{s}, \boldsymbol{a})} \left[ \log p_\chi(\boldsymbol{a}|\boldsymbol{s}', \boldsymbol{s}) \right] - \log \pi_\phi(\boldsymbol{a}|\boldsymbol{s})$, which is short-hand notation for the empowerment-induced addition to the reward signal—compare to Equation (5). We then commence with the objective for value function learning:

$$\min_\psi \mathbb{E}_{\boldsymbol{s} \sim \mathcal{D}} \left[ \left( V_\psi(\boldsymbol{s}) - \mathbb{E}_{\pi_\phi(\boldsymbol{a}|\boldsymbol{s})} \left[ Q_\theta(\boldsymbol{s}, \boldsymbol{a}) + \beta f(\boldsymbol{s}, \boldsymbol{a}) \right] \right)^2 \right], \qquad (17)$$

which is similar to the standard value objective but with the added term $\beta f(\boldsymbol{s}, \boldsymbol{a})$ as a result of joint cumulative empowerment learning. At this point, the necessity for a transition model $\mathcal{P}_\xi$ becomes apparent. In the above equation, new actions $\boldsymbol{a}$ need to be sampled from the policy $\pi_\phi$ for a given $\boldsymbol{s}$. However, the inverse dynamics model (inside $f$) depends on the subsequent state $\boldsymbol{s}'$ as well, requiring therefore a prediction for the next state. Note also that $(\boldsymbol{s}, \boldsymbol{a}, r, \boldsymbol{s}')$-tuples from the replay buffer as in Equation (16) can't be used here, because the expectation over $\boldsymbol{a}$ is w.r.t. to the current policy whereas tuples from the replay buffer come from a mixture of policies at an earlier stage of training.

Extending the ordinary actor-critic policy objective with the empowerment-induced term $f$ yields:

$$\max_\phi \mathbb{E}_{\boldsymbol{s} \sim \mathcal{D}} \left[ \mathbb{E}_{\pi_\phi(\boldsymbol{a}|\boldsymbol{s})} \left[ Q_\theta(\boldsymbol{s}, \boldsymbol{a}) + \beta f(\boldsymbol{s}, \boldsymbol{a}) \right] \right]. \qquad (18)$$

The remaining parameters to be optimized are $\chi$ and $\xi$ from the inverse dynamics model $p_\chi$ and the transition model $\mathcal{P}_\xi$. Both problems are supervised learning problems that can be addressed by log-likelihood maximization using samples from the replay buffer, leading to $\max_\chi \mathbb{E}_{\boldsymbol{s} \sim \mathcal{D}} \left[ \mathbb{E}_{\pi_\phi(\boldsymbol{a}|\boldsymbol{s}) \mathcal{P}_\xi(\boldsymbol{s}'|\boldsymbol{s}, \boldsymbol{a})} \left[ \log p_\chi(\boldsymbol{a}|\boldsymbol{s}', \boldsymbol{s}) \right] \right]$ and $\max_\xi \mathbb{E}_{\boldsymbol{s}, \boldsymbol{a}, \boldsymbol{s}' \sim \mathcal{D}} \left[ \log \mathcal{P}_\xi(\boldsymbol{s}'|\boldsymbol{s}, \boldsymbol{a}) \right]$.

Coming back to our motivation from Section 3, we propose two novel empowerment-inspired actor-critic approaches based on the optimization objectives specified in this section. The first combines cumulative reward maximization and empowerment learning following Equation (5) which we refer to as empowered actor-critic. The second learns cumulative empowerment values to construct intrinsic rewards following Equation (6) which we refer to as actor-critic with intrinsic empowerment.

**Empowered Actor-Critic (EAC).** In line with standard off-policy actor-critic methods [36, 15, 21], EAC interacts with the environment iteratively storing transition tuples $(\boldsymbol{s}, \boldsymbol{a}, r, \boldsymbol{s}')$ in a replay buffer. After each interaction, a training batch $\{(\boldsymbol{s}, \boldsymbol{a}, r, \boldsymbol{s}')^{(b)}\}_{b=1}^B \sim \mathcal{D}$ of size $B$ is sampled from the buffer to perform a *single* gradient update on the objectives from Equation (16) to (18) as well as the log likelihood objectives for the inverse dynamics and transition model—see Appendix B for pseudocode.

**Actor-Critic with Intrinsic Empowerment (ACIE).** By setting $\alpha = 0$ and $\beta = 1$, EAC can train an agent merely focusing on cumulative empowerment learning. Since EAC is off-policy, it can learn with samples obtained from executing *any* policy in the real environment, e.g. the actor of *any other* reward-maximizing actor-critic algorithm. We can then extend external rewards $r_t$ at time $t$ of this actor-critic algorithm with intrinsic rewards $\mathbb{E}_{\pi_\phi(\boldsymbol{a}|\boldsymbol{s}_t) \mathcal{P}_\xi(\boldsymbol{s}'|\boldsymbol{s}_t, \boldsymbol{a})} \left[ \log \frac{p_\chi(\boldsymbol{a}|\boldsymbol{s}', \boldsymbol{s}_t)}{\pi_\phi(\boldsymbol{a}|\boldsymbol{s}_t)} \right]$ according to Equation (6), where $(\phi, \xi, \chi)$ are the result of *concurrent* raw empowerment learning with EAC. This idea is similar to the preliminary work of [29] using non-cumulative empowerment as intrinsic motivation for deep value-based RL with discrete actions in the Atari game Montezuma's Revenge.

## 5.2 Experiments with Deep Function Approximators in MuJoCo

We validate EAC and ACIE in the robotics simulator MuJoCo [66, 8] with deep neural nets under the same setup for each experiment following [67, 25, 50, 24, 56, 36, 68, 57, 1, 9, 15, 21]—see Appendix C.2 for details. While EAC is a standalone algorithm, ACIE can be combined with any RL algorithm (we use the model-free state of the art SAC [21]). We compare against DDPG [36] and PPO [57] from RLlib [35] as well as SAC on the MuJoCo v2-environments (ten seeds per run [47]).

The results in Figure 2 confirm that both EAC and ACIE can attain better initial performance compared to model-free baselines. While this holds true for both approaches on the pendulum benchmarks (balancing and swing up), our empowered methods can also boost RL in demanding environments like Hopper, Ant and Humanoid (the latter two being amongst the most difficult MuJoCo tasks). EAC significantly improves initial learning in Ant, whereas ACIE boosts SAC in Hopper and Humanoid. While EAC outperforms PPO and DDPG in almost all tasks, it is not consistently better then SAC. Similarly, the added intrinsic reward from ACIE to SAC does not always help. *This is not unexpected as it cannot be in general ruled out that reward functions assign high (low) rewards to lowly (highly) empowered states, in which case the two learning signals may become partially conflicting.*

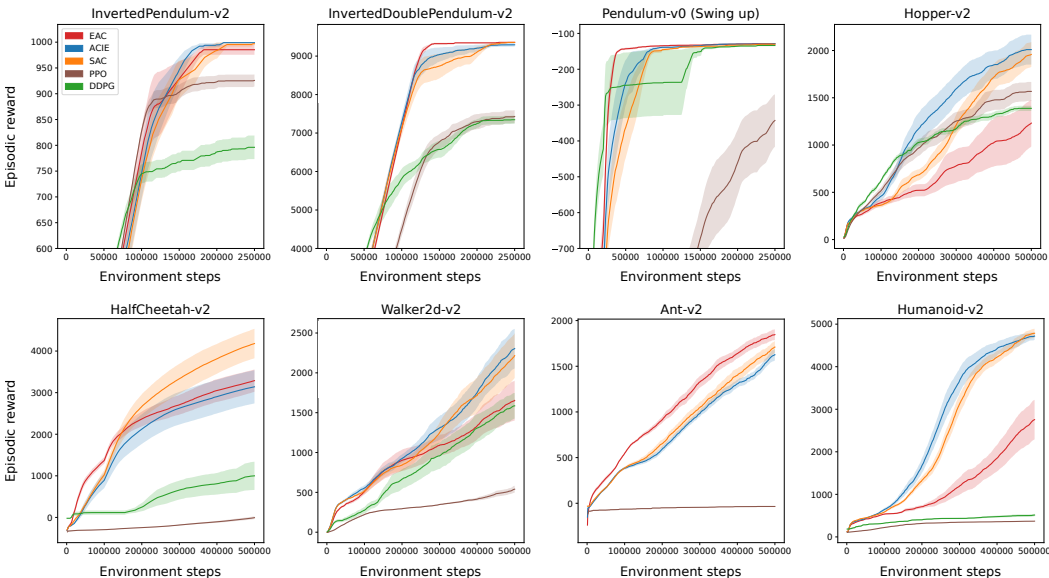

Figure 2: MuJoCo Experiments. The plots show maximum episodic rewards (averaged over the last 100 episodes) achieved so far [9] versus steps—*non-maximum* episodic reward plots can be found in Figure 3. EAC and ACIE are compared to DDPG, PPO and SAC (DDPG did not work in Ant, see [21] and Appendix C.2 for an explanation). Shaded areas refer to the standard error. Both EAC and ACIE improve initial learning over baselines in the three pendulum tasks (upper row). In demanding problems like Hopper, Ant and Humanoid, our methods can boost RL. In terms of final performance, EAC is competitive with the baselines: it consistently outperforms DDPG and PPO on all tasks except Hopper, but is not always better than SAC. Similarly, the ACIE-signal does not always help SAC. This is not unexpected as extrinsic and empowered rewards may partially conflict.

For the sake of completeness, we report Figure 3 which is similar to Figure 2 but shows episodic rewards and *not* maximum episodic rewards obtained so far [9]. Also, limits of y-axes are preserved for the pendulum tasks. Note that our SAC baseline is comparable with the SAC from [22] on Hopper-v2, Walker2d-v2, Ant-v2 and Humanoid-v2 after $5 \cdot 10^5$ steps (the SAC from [21] uses the earlier v1-versions of Mujoco and is hence not an optimal reference). However, there is a discrepancy on HalfCheetah-v2. This was earlier noted by others who tried to reproduce SAC results in HalfCheetah-v2 but failed to obtain episodic rewards as high as in [21, 22], leading to a GitHub issue https://github.com/rail-berkeley/softlearning/issues/75. The final conclusion of this issue was that differences in performance are caused by different seed settings and are therefore of statistical nature (comparing all algorithms under the same seed settings is hence valid).

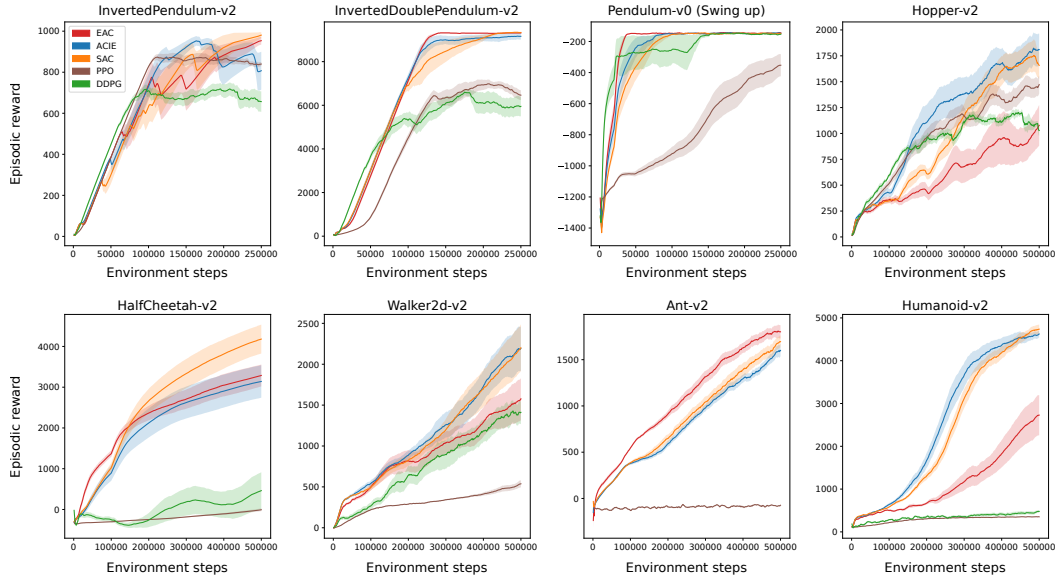

Figure 3: Raw Results of MuJoCo Experiments. The plots are similar to the plots from Figure 2, but report episodic rewards (averaged over the last 100 episodes) versus steps—*not* maximum episodic rewards seen so far as in [9]. For the pendulum tasks, the limits of the y-axes are preserved.

# 6 Conclusion

This paper provides a theoretical contribution via a unified formulation for reward maximization and empowerment that generalizes Bellman's optimality principle and recent information-theoretic extensions to it. We proved the existence of and convergence to unique optimal values, and practically validated our ideas by devising novel parametric actor-critic algorithms inspired by our formulation. These were evaluated on the high-dimensional MuJoCo benchmark demonstrating that empowerment can boost RL in challenging robotics tasks (e.g. Ant and Humanoid).

The most promising line of future research is to investigate scheduling schemes that dynamically trade off rewards vs. empowerment with the prospect of obtaining better asymptotic performance. Empowerment could also be particularly useful in a multi-task setting where task transfer could benefit from initially empowered agents.

### Acknowledgments

We thank Haitham Bou-Ammar for pointing us in the direction of empowerment.

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
