[Supplementary Material]

# A  Theoretical Analysis

This section provides more details regarding the theoretical analysis of the main paper to prove the existence of unique optimal values as well as convergence of the value iteration scheme.

## A.1   Proof of Proposition 2 from the Main Paper

**Proof.** Following [5, 51, 18], let's start by defining $P_{\pi_{\text{behave}}} : \mathcal{S} \times \mathcal{S} \to [0, 1]$ and $g_{q, \pi_{\text{behave}}} : \mathcal{S} \to \mathbb{R}$:

$$P_{\pi_{\text{behave}}}(\boldsymbol{s}, \boldsymbol{s}') := \mathbb{E}_{\pi_{\text{behave}}(\boldsymbol{a}|\boldsymbol{s})} \left[ \mathcal{P}(\boldsymbol{s}'|\boldsymbol{s}, \boldsymbol{a}) \right],$$

$$g_{q, \pi_{\text{behave}}}(\boldsymbol{s}) := \mathbb{E}_{\pi_{\text{behave}}(\boldsymbol{a}|\boldsymbol{s})} \left[ \alpha \mathcal{R}(\boldsymbol{s}, \boldsymbol{a}) + \beta \mathbb{E}_{\mathcal{P}(\boldsymbol{s}'|\boldsymbol{s}, \boldsymbol{a})} \left[ \log \frac{q(\boldsymbol{a}|\boldsymbol{s}', \boldsymbol{s})}{\pi_{\text{behave}}(\boldsymbol{a}|\boldsymbol{s})} \right] \right].$$

We can then express the Bellman operator $B_{q, \pi_{\text{behave}}}$ in vectorized form yielding $B_{q, \pi_{\text{behave}}} V = g_{q, \pi_{\text{behave}}} + \gamma P_{\pi_{\text{behave}}} V$. Defining $B_{q, \pi_{\text{behave}}}^{(i)}$ as short-hand notation for applying $B_{q, \pi_{\text{behave}}}$ to a value vector $V$ $i$-times consecutively ($i = 0$ leaves $V$ unaffected), we arrive at:

$$V^{(q, \pi_{\text{behave}})} := \lim_{i \to \infty} B_{q, \pi_{\text{behave}}}^{(i)} V = \lim_{i \to \infty} \sum_{t=0}^{i-1} \gamma^t P_{\pi_{\text{behave}}}^t g_{q, \pi_{\text{behave}}} + \underbrace{\gamma^i P_{\pi_{\text{behave}}}^i V}_{\to 0},$$

where $P_{\pi_{\text{behave}}}^t$ denotes the $t$-times multiplication of $P_{\pi_{\text{behave}}}$ with itself ($P_{\pi_{\text{behave}}}^0$ is the identity matrix). This means that the convergence of $B_{q, \pi_{\text{behave}}}$ does not depend on the initial value vector $V$, therefore:

$$B_{q, \pi_{\text{behave}}} V^{(q, \pi_{\text{behave}})} = g_{q, \pi_{\text{behave}}} + \gamma P_{\pi_{\text{behave}}} \lim_{i \to \infty} \sum_{t=0}^{i-1} \gamma^t P_{\pi_{\text{behave}}}^t g_{q, \pi_{\text{behave}}}$$

$$= \gamma^0 P_{\pi_{\text{behave}}}^0 g_{q, \pi_{\text{behave}}} + \lim_{i \to \infty} \sum_{t=1}^{i} \gamma^t P_{\pi_{\text{behave}}}^t g_{q, \pi_{\text{behave}}}$$

$$= \lim_{i \to \infty} \sum_{t=0}^{i-1} \gamma^t P_{\pi_{\text{behave}}}^t g_{q, \pi_{\text{behave}}} + \underbrace{\gamma^i P_{\pi_{\text{behave}}}^i g_{q, \pi_{\text{behave}}}}_{\to 0} = V^{(q, \pi_{\text{behave}})},$$

proving that $V^{(q, \pi_{\text{behave}})}$ is a fixed point of $B_{q, \pi_{\text{behave}}}$. The uniqueness proof follows next. Assume there was another fixed point $V'$ of $B_{q, \pi_{\text{behave}}}$, then $\lim_{i \to \infty} B_{q, \pi_{\text{behave}}}^{(i)} V' = V^{(q, \pi_{\text{behave}})}$ because the convergence behavior of $B_{q, \pi_{\text{behave}}}$ does not depend on the initial $V'$, hence $V' = V^{(q, \pi_{\text{behave}})}$. $\square$

## A.2   Proof of Proposition 3 from the Main Paper

**Proof.** Proving Proposition 3 from the main paper is similar to the maximum channel capacity problem from information theory [59, 10, 16]. The proof follows hence similar steps as the one for Proposition 1 from the background section on empowerment in the main paper, in the following accomplished via Lemma 1, 2 and 3. $\square$

**Lemma 1** *Inverse Dynamics. Maximizing the right-hand side of the Bellman operator $B_\star V(\boldsymbol{s}) = \max_{\pi_{\text{behave}}, q} B_{q, \pi_{\text{behave}}} V(\boldsymbol{s})$ w.r.t. to $q$ for a given $\pi_{\text{behave}}$ yields:*

$$\operatorname{argmax}_q B_{q, \pi_{\text{behave}}} V(\boldsymbol{s}) = \frac{\mathcal{P}(\boldsymbol{s}'|\boldsymbol{s}, \boldsymbol{a}) \pi_{\text{behave}}(\boldsymbol{a}|\boldsymbol{s})}{\sum_{\boldsymbol{a}} \mathcal{P}(\boldsymbol{s}'|\boldsymbol{s}, \boldsymbol{a}) \pi_{\text{behave}}(\boldsymbol{a}|\boldsymbol{s})}.$$

**Proof.** It holds that $\operatorname{argmax}_q B_{q, \pi_{\text{behave}}} V(\boldsymbol{s}) = \operatorname{argmax}_q \mathbb{E}_{\pi_{\text{behave}}(\boldsymbol{a}|\boldsymbol{s}) \mathcal{P}(\boldsymbol{s}'|\boldsymbol{s}, \boldsymbol{a})} \left[ \log \frac{q(\boldsymbol{a}|\boldsymbol{s}', \boldsymbol{s})}{\pi_{\text{behave}}(\boldsymbol{a}|\boldsymbol{s})} \right]$ because neither $\mathcal{R}$ nor $V$ depends on $q$. It then follows that

$$\mathbb{E}_{\pi_{\text{behave}}(\boldsymbol{a}|\boldsymbol{s}) \mathcal{P}(\boldsymbol{s}'|\boldsymbol{s}, \boldsymbol{a})} \left[ \log \frac{q(\boldsymbol{a}|\boldsymbol{s}', \boldsymbol{s})}{\pi_{\text{behave}}(\boldsymbol{a}|\boldsymbol{s})} \right] \overset{\forall q}{\leq} I(\boldsymbol{A}, \boldsymbol{S}'|\boldsymbol{s}) = \mathbb{E}_{\pi_{\text{behave}}(\boldsymbol{a}|\boldsymbol{s}) \mathcal{P}(\boldsymbol{s}'|\boldsymbol{s}, \boldsymbol{a})} \left[ \log \frac{p(\boldsymbol{a}|\boldsymbol{s}', \boldsymbol{s})}{\pi_{\text{behave}}(\boldsymbol{a}|\boldsymbol{s})} \right],$$

where $p$ is the true Bayesian posterior—see [10] Lemma 10.8.1. $\square$

**Lemma 2** *Optimal Policy. Maximizing the right-hand side of the Bellman operator* $B_\star V(\boldsymbol{s}) = \max_{\pi_{\text{behave}},q} B_{q,\pi_{\text{behave}}} V(\boldsymbol{s})$ *w.r.t. to* $\pi_{\text{behave}}$ *for a given q yields:*

$$\operatorname{argmax}_{\pi_{\text{behave}}} B_{q,\pi_{\text{behave}}} V(\boldsymbol{s}) = \frac{\exp\left(\frac{\alpha}{\beta}\mathcal{R}(\boldsymbol{s},\boldsymbol{a}) + \mathbb{E}_{\mathcal{P}(\boldsymbol{s'}|\boldsymbol{s},\boldsymbol{a})}\left[\log q(\boldsymbol{a}|\boldsymbol{s'},\boldsymbol{s}) + \frac{\gamma}{\beta}V(\boldsymbol{s'})\right]\right)}{\sum_{\boldsymbol{a}} \exp\left(\frac{\alpha}{\beta}\mathcal{R}(\boldsymbol{s},\boldsymbol{a}) + \mathbb{E}_{\mathcal{P}(\boldsymbol{s'}|\boldsymbol{s},\boldsymbol{a})}\left[\log q(\boldsymbol{a}|\boldsymbol{s'},\boldsymbol{s}) + \frac{\gamma}{\beta}V(\boldsymbol{s'})\right]\right)}.$$

**Proof.** Maximizing $B_{q,\pi_{\text{behave}}} V(\boldsymbol{s})$ w.r.t. $\pi_{\text{behave}}$ subject to the constraint $\sum_{\boldsymbol{a}} \pi_{\text{behave}}(\boldsymbol{a}|\boldsymbol{s}) = 1$ yields the Lagrangian:

$$L(\pi_{\text{behave}},\lambda) = B_{q,\pi_{\text{behave}}} V(\boldsymbol{s}) - \lambda\left(\left(\sum_{\boldsymbol{a}} \pi_{\text{behave}}(\boldsymbol{a}|\boldsymbol{s})\right) - 1\right),$$

where $\lambda$ is a Lagrange multiplier. The derivatives of the Lagrangian w.r.t. $\pi_{\text{behave}}(\tilde{\boldsymbol{a}}|\boldsymbol{s})$, where $\tilde{\boldsymbol{a}}$ refers to a specific action, and $\lambda$ are given by:

$$\frac{\partial L(\pi_{\text{behave}},\lambda)}{\partial \pi_{\text{behave}}(\tilde{\boldsymbol{a}}|\boldsymbol{s})} = \alpha\mathcal{R}(\boldsymbol{s},\tilde{\boldsymbol{a}}) + \mathbb{E}_{\mathcal{P}(\boldsymbol{s'}|\boldsymbol{s},\tilde{\boldsymbol{a}})}\left[\beta\log\frac{q(\tilde{\boldsymbol{a}}|\boldsymbol{s'},\boldsymbol{s})}{\pi_{\text{behave}}(\tilde{\boldsymbol{a}}|\boldsymbol{s})} + \gamma V(\boldsymbol{s'})\right] - \beta - \lambda,$$

$$\frac{\partial L(\pi_{\text{behave}},\lambda)}{\partial \lambda} = -\left(\left(\sum_{\boldsymbol{a}} \pi_{\text{behave}}(\boldsymbol{a}|\boldsymbol{s})\right) - 1\right).$$

Equating the first derivative with 0 and resolving w.r.t. $\pi_{\text{behave}}(\tilde{\boldsymbol{a}}|\boldsymbol{s})$, one arrives at:

$$\pi_{\text{behave}}(\tilde{\boldsymbol{a}}|\boldsymbol{s}) = \exp\left(\frac{\alpha}{\beta}\mathcal{R}(\boldsymbol{s},\tilde{\boldsymbol{a}}) + \mathbb{E}_{\mathcal{P}(\boldsymbol{s'}|\boldsymbol{s},\tilde{\boldsymbol{a}})}\left[\log q(\tilde{\boldsymbol{a}}|\boldsymbol{s'},\boldsymbol{s}) + \frac{\gamma}{\beta}V(\boldsymbol{s'})\right] - \frac{\beta+\lambda}{\beta}\right).$$

Plugging this result into the second derivative and equating with 0 yields:

$$\exp\left(-\frac{\beta+\lambda}{\beta}\right) = \left(\sum_{\boldsymbol{a}} \exp\left(\frac{\alpha}{\beta}\mathcal{R}(\boldsymbol{s},\boldsymbol{a}) + \mathbb{E}_{\mathcal{P}(\boldsymbol{s'}|\boldsymbol{s},\boldsymbol{a})}\left[\log q(\boldsymbol{a}|\boldsymbol{s'},\boldsymbol{s}) + \frac{\gamma}{\beta}V(\boldsymbol{s'})\right]\right)\right)^{-1}.$$

Plugging the latter back into the result for $\pi_{\text{behave}}(\tilde{\boldsymbol{a}}|\boldsymbol{s})$ completes the proof. $\qquad\square$

**Lemma 3** *Blahut-Arimoto. Assuming bounded $\mathcal{R}$, iterating through Equations (13) and (14) from the main paper converges to* $\operatorname{argmax}_{\pi_{\text{behave}},q} B_{q,\pi_{\text{behave}}} V(\boldsymbol{s})$ *at a rate of $\mathcal{O}(1/M)$ for arbitrary initial $\pi_{\text{behave}}^{(0)}$ having support in $\mathcal{A}$ $\forall \boldsymbol{s}$, with $M$ being the total number of iterations.*

**Proof.** Evaluating the operator $B_{q,\pi_{\text{behave}}} V(\boldsymbol{s})$ at the pair $(q^{(m)}, \pi_{\text{behave}}^{(m+1)})$, we obtain:

$$B_{q^{(m)},\pi_{\text{behave}}^{(m+1)}} V(\boldsymbol{s}) = \beta\log\sum_{\boldsymbol{a}} \exp\left(\frac{\alpha}{\beta}\mathcal{R}(\boldsymbol{s},\boldsymbol{a}) + \mathbb{E}_{\mathcal{P}(\boldsymbol{s'}|\boldsymbol{s},\boldsymbol{a})}\left[\log q^{(m)}(\boldsymbol{a}|\boldsymbol{s'},\boldsymbol{s}) + \frac{\gamma}{\beta}V(\boldsymbol{s'})\right]\right).$$

Due to Lemma 4, we know that $\max_{\pi_{\text{behave}},q} B_{q,\pi_{\text{behave}}} V(\boldsymbol{s})$ is upper bounded:

$$\max_{\pi_{\text{behave}},q} B_{q,\pi_{\text{behave}}} V(\boldsymbol{s}) \leq$$

$$\mathbb{E}_{\pi_{\text{behave}}^{`\star`}(\boldsymbol{a}|\boldsymbol{s})}\left[\alpha\mathcal{R}(\boldsymbol{s},\boldsymbol{a}) + \mathbb{E}_{\mathcal{P}(\boldsymbol{s'}|\boldsymbol{s},\boldsymbol{a})}\left[\beta\log q^{(m)}(\boldsymbol{a}|\boldsymbol{s'},\boldsymbol{s}) + \gamma V(\boldsymbol{s'})\right] - \beta\log\pi_{\text{behave}}^{(m)}(\boldsymbol{a}|\boldsymbol{s})\right] =$$

$$\mathbb{E}_{\pi_{\text{behave}}^{`\star`}(\boldsymbol{a}|\boldsymbol{s})}\left[\beta\log\left(\exp\left(\frac{\alpha}{\beta}\mathcal{R}(\boldsymbol{s},\boldsymbol{a}) + \mathbb{E}_{\mathcal{P}(\boldsymbol{s'}|\boldsymbol{s},\boldsymbol{a})}\left[\log q^{(m)}(\boldsymbol{a}|\boldsymbol{s'},\boldsymbol{s}) + \frac{\gamma}{\beta}V(\boldsymbol{s'})\right]\right)\right) - \beta\log\pi_{\text{behave}}^{(m)}(\boldsymbol{a}|\boldsymbol{s})\right],$$

where the notation `$\star$` indicates optimality of a single value iteration step, as opposed to the notation $(q^\star, \pi_{\text{behave}}^\star)$ from the main paper that refers to optimality after the entire value iteration scheme has converged—see Lemma 4.

By using the definition of $\pi_{\text{behave}}^{(m+1)}(\boldsymbol{a}|\boldsymbol{s})$ from Equation (14), the upper two equations enable us to derive the following upper bound:

$$\max_{\pi_{\text{behave}},q} B_{q,\pi_{\text{behave}}} V(\boldsymbol{s}) - B_{q^{(m)},\pi_{\text{behave}}^{(m+1)}} V(\boldsymbol{s}) \leq \beta\mathbb{E}_{\pi_{\text{behave}}^{`\star`}(\boldsymbol{a}|\boldsymbol{s})}\left[\log\frac{\pi_{\text{behave}}^{(m+1)}(\boldsymbol{a}|\boldsymbol{s})}{\pi_{\text{behave}}^{(m)}(\boldsymbol{a}|\boldsymbol{s})}\right].$$

From there it follows that for $M$ steps of the Blahut-Arimoto scheme

$$\frac{1}{M}\sum_{m=0}^{M-1}\left(\max_{\pi_{\text{behave}},q} B_{q,\pi_{\text{behave}}}V(\boldsymbol{s}) - B_{q^{(m)},\pi_{\text{behave}}^{(m+1)}}V(\boldsymbol{s})\right) \leq \frac{1}{M}\beta\mathbb{E}_{\pi_{\text{behave}}^{`\star`}(\boldsymbol{a}|\boldsymbol{s})}\left[\log\frac{\pi_{\text{behave}}^{(M)}(\boldsymbol{a}|\boldsymbol{s})}{\pi_{\text{behave}}^{(0)}(\boldsymbol{a}|\boldsymbol{s})}\right] \leq$$

$$\frac{1}{M}\beta\mathbb{E}_{\pi_{\text{behave}}^{`\star`}(\boldsymbol{a}|\boldsymbol{s})}\left[\log\frac{1}{\pi_{\text{behave}}^{(0)}(\boldsymbol{a}|\boldsymbol{s})}\right] \leq \frac{1}{M}\beta\max_{\boldsymbol{a}}\left[\log\frac{1}{\pi_{\text{behave}}^{(0)}(\boldsymbol{a}|\boldsymbol{s})}\right].$$

However, since the upper term is lower-bounded by 0 and since $B_{q^{(0)},\pi_{\text{behave}}^{(0)}}V(\boldsymbol{s}) \leq B_{q^{(0)},\pi_{\text{behave}}^{(1)}}V(\boldsymbol{s}) \leq B_{q^{(1)},\pi_{\text{behave}}^{(1)}}V(\boldsymbol{s}) \leq \dots$ because of the alternating optimization procedure, this implies convergence at a rate of $\mathcal{O}(1/M)$. $\square$

**Lemma 4** *Upper Value Bound for One Value Iteration Step. Let's introduce the following notation* $(q^{`\star`}, \pi_{\text{behave}}^{`\star`}) := \text{argmax}_{\pi_{\text{behave}},q} B_{q,\pi_{\text{behave}}}V(\boldsymbol{s})$ *where the symbol '$\star$' indicates optimality of a single value iteration step, as opposed to the notation* $(q^{\star}, \pi_{\text{behave}}^{\star})$ *from the main paper that refers to optimality after the entire value iteration scheme has converged. Let's define* $\kappa^{(m)}(\boldsymbol{s},\boldsymbol{a}) := \alpha\mathcal{R}(\boldsymbol{s},\boldsymbol{a}) + \mathbb{E}_{\mathcal{P}(\boldsymbol{s}'|\boldsymbol{s},\boldsymbol{a})}\left[\beta\log q^{(m)}(\boldsymbol{a}|\boldsymbol{s}',\boldsymbol{s}) + \gamma V(\boldsymbol{s}')\right]$. *It then holds that* $\max_{\pi_{\text{behave}},q} B_{q,\pi_{\text{behave}}}V(\boldsymbol{s}) \leq \mathbb{E}_{\pi_{\text{behave}}^{`\star`}(\boldsymbol{a}|\boldsymbol{s})}\left[\kappa^{(m)}(\boldsymbol{s},\boldsymbol{a}) - \beta\log\pi_{\text{behave}}^{(m)}(\boldsymbol{a}|\boldsymbol{s})\right]$.

**Proof.** Let's first note that $(q^{`\star`}, \pi_{\text{behave}}^{`\star`})$ exists because $B_{q,\pi_{\text{behave}}}V$ is bounded. $B_{q,\pi_{\text{behave}}}V$ is bounded because it is a sum of three weighted terms that are bounded—see Equation (12) of the main paper:

- $\mathbb{E}_{\pi_{\text{behave}}(\boldsymbol{a}|\boldsymbol{s})}\left[\mathcal{R}(\boldsymbol{s},\boldsymbol{a})\right]$ is bounded because the reward is bounded by assumption,
- $\mathbb{E}_{\pi_{\text{behave}}(\boldsymbol{a}|\boldsymbol{s})\mathcal{P}(\boldsymbol{s}'|\boldsymbol{s},\boldsymbol{a})}\left[\log\frac{q(\boldsymbol{a}|\boldsymbol{s}',\boldsymbol{s})}{\pi_{\text{behave}}(\boldsymbol{a}|\boldsymbol{s})}\right]$ is a lower bound to the mutual information $I(\boldsymbol{A},\boldsymbol{S}'|\boldsymbol{s})$ (which is bounded) according to [10] Lemma 10.8.1,
- and $V(\boldsymbol{s}')$ is bounded when the value iteration schemes (both using $B_{\star}$ and $B_{q,\pi_{\text{behave}}}$) are initialized, and remains bounded in each value iteration step because $B_{q,\pi_{\text{behave}}}V(\boldsymbol{s})$ is bounded due to the previous two points and initial bounded $V(\boldsymbol{s})$.

It then holds that

$$\max_{\pi_{\text{behave}},q} B_{q,\pi_{\text{behave}}}V(\boldsymbol{s}) = B_{q^{`\star`},\pi_{\text{behave}}^{`\star`}}V(\boldsymbol{s})$$

$$= \mathbb{E}_{\pi_{\text{behave}}^{`\star`}(\boldsymbol{a}|\boldsymbol{s})}\left[\alpha\mathcal{R}(\boldsymbol{s},\boldsymbol{a}) + \gamma\mathbb{E}_{\mathcal{P}(\boldsymbol{s}'|\boldsymbol{s},\boldsymbol{a})}\left[V(\boldsymbol{s}')\right] + \beta\mathbb{E}_{\mathcal{P}(\boldsymbol{s}'|\boldsymbol{s},\boldsymbol{a})}\left[\log\frac{\mathcal{P}(\boldsymbol{s}'|\boldsymbol{s},\boldsymbol{a})}{\sum_{\boldsymbol{a}}\mathcal{P}(\boldsymbol{s}'|\boldsymbol{s},\boldsymbol{a})\pi_{\text{behave}}^{`\star`}(\boldsymbol{a}|\boldsymbol{s})}\right]\right]$$

$$\leq \mathbb{E}_{\pi_{\text{behave}}^{`\star`}(\boldsymbol{a}|\boldsymbol{s})}\left[\alpha\mathcal{R}(\boldsymbol{s},\boldsymbol{a}) + \gamma\mathbb{E}_{\mathcal{P}(\boldsymbol{s}'|\boldsymbol{s},\boldsymbol{a})}\left[V(\boldsymbol{s}')\right] + \beta\mathbb{E}_{\mathcal{P}(\boldsymbol{s}'|\boldsymbol{s},\boldsymbol{a})}\left[\log\frac{\mathcal{P}(\boldsymbol{s}'|\boldsymbol{s},\boldsymbol{a})}{\sum_{\boldsymbol{a}}\mathcal{P}(\boldsymbol{s}'|\boldsymbol{s},\boldsymbol{a})\pi_{\text{behave}}^{(m)}(\boldsymbol{a}|\boldsymbol{s})}\right]\right],$$

where the equality is obtained by plugging in $q^{`\star`}$ using Equation (13), and where the inequality leverages one more time [10] Lemma 10.8.1.

At the same time, we can plug Equation (13) from the main paper into $\kappa^{(m)}(\boldsymbol{s},\boldsymbol{a})$, yielding:

$$\kappa^{(m)}(\boldsymbol{s},\boldsymbol{a}) = \alpha\mathcal{R}(\boldsymbol{s},\boldsymbol{a}) + \mathbb{E}_{\mathcal{P}(\boldsymbol{s}'|\boldsymbol{s},\boldsymbol{a})}\left[\beta\log\frac{\mathcal{P}(\boldsymbol{s}'|\boldsymbol{s},\boldsymbol{a})\pi_{\text{behave}}^{(m)}(\boldsymbol{a}|\boldsymbol{s})}{\sum_{\boldsymbol{a}}\mathcal{P}(\boldsymbol{s}'|\boldsymbol{s},\boldsymbol{a})\pi_{\text{behave}}^{(m)}(\boldsymbol{a}|\boldsymbol{s})} + \gamma V(\boldsymbol{s}')\right].$$

Rearranging the upper equation results in:

$$\beta\mathbb{E}_{\mathcal{P}(\boldsymbol{s}'|\boldsymbol{s},\boldsymbol{a})}\left[\log\frac{\mathcal{P}(\boldsymbol{s}'|\boldsymbol{s},\boldsymbol{a})}{\sum_{\boldsymbol{a}}\mathcal{P}(\boldsymbol{s}'|\boldsymbol{s},\boldsymbol{a})\pi_{\text{behave}}^{(m)}(\boldsymbol{a}|\boldsymbol{s})}\right] =$$

$$\kappa^{(m)}(\boldsymbol{s},\boldsymbol{a}) - \beta\log\pi_{\text{behave}}^{(m)}(\boldsymbol{a}|\boldsymbol{s}) - \alpha\mathcal{R}(\boldsymbol{s},\boldsymbol{a}) - \gamma\mathbb{E}_{\mathcal{P}(\boldsymbol{s}'|\boldsymbol{s},\boldsymbol{a})}\left[V(\boldsymbol{s}')\right].$$

Plugging the latter result into the earlier derived inequality completes the proof. $\square$

## A.3 Proof of Proposition 4 from the Main Paper

**Proof.** The mechanics of the proof are in line with [5, 51, 18]. Let's denote $(q^\star, \pi_{\text{behave}}^\star) = \text{argmax}_{\pi_{\text{behave}},q} V^{(q,\pi_{\text{behave}})}$ and $V^\star = V^{(q^\star,\pi_{\text{behave}}^\star)}$. It then holds that

$$V^\star = B_{q^\star,\pi_{\text{behave}}^\star} V^\star \leq \max_{\pi_{\text{behave}},q} B_{q,\pi_{\text{behave}}} V^\star =: B_{q',\pi_{\text{behave}}'} V^\star \leq V^{(q',\pi_{\text{behave}}')},$$

where the last inequality is because of the consistency of values as proven in Lemma 5. But by definition it holds that $V^\star = \max_{\pi_{\text{behave}},q} V^{(q,\pi_{\text{behave}})} \geq V^{(q',\pi_{\text{behave}}')}$. This implies that $V^\star = V^{(q',\pi_{\text{behave}}')}$. The latter means that $V^\star = \max_{\pi_{\text{behave}},q} B_{q,\pi_{\text{behave}}} V^\star = B_\star V^\star$ which proves that $V^\star$ is a fixed point of the operator $B_\star$.

The uniqueness of values proof comes next. Assume there was another fixed point of the operator $B_\star$ denoted as $V' = V^{(q',\pi_{\text{behave}}')}$, then

$$V^\star = B_{q^\star,\pi_{\text{behave}}^\star} V^\star = \max_{\pi_{\text{behave}},q} B_{q,\pi_{\text{behave}}} V^\star \geq B_{q',\pi_{\text{behave}}'} V^\star \geq V^{(q',\pi_{\text{behave}}')} = V',$$

where the last inequality is again because of Lemma 5. One can show similarly that $V' \geq V^\star$, which does hence imply that $V' = V^\star$. $\qquad\square$

**Lemma 5** *Value Consistency for the Evaluation Operator. If $V \leq B_{q,\pi_{\text{behave}}} V$ then $B_{q,\pi_{\text{behave}}}^{(i)} V \leq V^{(q,\pi_{\text{behave}})}$ $\forall i \in \mathbb{N}$, and similarly if $V \geq B_{q,\pi_{\text{behave}}} V$ then $B_{q,\pi_{\text{behave}}}^{(i)} V \geq V^{(q,\pi_{\text{behave}})}$ $\forall i \in \mathbb{N}$.*

**Proof.** The proof follows via induction. The base case is $V \overset{(\geq)}{\leq} B_{q,\pi_{\text{behave}}} V$. The inductive step is as follows. If $B_{q,\pi_{\text{behave}}}^{(i-1)} V \overset{(\geq)}{\leq} B_{q,\pi_{\text{behave}}}^{(i)} V$ then

$$B_{q,\pi_{\text{behave}}}^{(i+1)} V = g_{q,\pi_{\text{behave}}} + \gamma P_{\pi_{\text{behave}}} B_{q,\pi_{\text{behave}}}^{(i)} V \overset{(\leq)}{\geq} g_{q,\pi_{\text{behave}}} + \gamma P_{\pi_{\text{behave}}} B_{q,\pi_{\text{behave}}}^{(i-1)} V = B_{q,\pi_{\text{behave}}}^{(i)} V,$$

which completes the induction with help of the concise notation from Appendix A.1. $\qquad\square$

## A.4 Proof Details of Theorem 2 from the Main Paper

This section is to shed more light on the proof of Theorem 2 from the main paper to show that $B_\star$ is a contraction map via the subsequent proposition.

**Proposition 5** *Contraction Map. Assuming bounded $\mathcal{R}$ and let $\eta \in \mathbb{R}^+$ be a positive constant $\eta = \alpha \max_{s,a} |\mathcal{R}(s,a)| + \beta \log |\mathcal{A}|$. Then $\left\| V^\star - B_\star^{(i)} V \right\|_\infty \leq \gamma^i \frac{1}{1-\gamma} \eta$ with initial $V(s) = 0$ $\forall s$.*

**Proof.** The proposition is proven by the following sequence of inequalities:

$$\left\| V^\star - B_\star^{(i)} V \right\|_\infty =: \left| V^\star(s^\star) - B_\star^{(i)} V(s^\star) \right| =$$

$$\left| \max_{\pi_{\text{behave}},q} B_{q,\pi_{\text{behave}}} V^\star(s^\star) - \max_{\pi_{\text{behave}},q} B_{q,\pi_{\text{behave}}} B_\star^{(i-1)} V(s^\star) \right| \leq$$

$$\max_{\pi_{\text{behave}},q} \left| B_{q,\pi_{\text{behave}}} V^\star(s^\star) - B_{q,\pi_{\text{behave}}} B_\star^{(i-1)} V(s^\star) \right| =$$

$$\max_{\pi_{\text{behave}}} \left| \gamma \mathbb{E}_{\pi_{\text{behave}}(a|s)\mathcal{P}(s'|s,a)} \left[ V^\star(s') \right] - \gamma \mathbb{E}_{\pi_{\text{behave}}(a|s)\mathcal{P}(s'|s,a)} \left[ B_\star^{(i-1)} V(s') \right] \right| \leq$$

$$\gamma \left\| V^\star - B_\star^{(i-1)} V \right\|_\infty \overset{\text{recursion}}{\leq} \gamma^i \left\| V^\star - V \right\|_\infty \overset{V \text{ is } 0}{=} \gamma^i \left\| V^\star \right\|_\infty \leq \gamma^i \frac{1}{1-\gamma} \eta,$$

where $\eta$ is a positive constant to upper-bound $V^\star$-values, see Corollary 2. $\qquad\square$

**Corollary 2** *Upper Value Bound for Optimal Values. Optimal values are upper-bounded according to $|V^\star(s)| \leq \frac{1}{1-\gamma} \left( \alpha \max_{s,a} |\mathcal{R}(s,a)| + \beta \log |\mathcal{A}| \right)$ $\forall s$.*

This follows straightforwardly from worst-case assumptions and properties of the geometric series and the mutual information. The empowerment-induced addition to the reward signal is upper-bounded by a mutual information term, which is upper-bounded by the worst-case entropy in action space.

**Remark.** *A contraction proof for $B_\star$ with any two initial value vectors $V'$ and $V$ follows similar steps as outlined in Proposition 5 by replacing $V^\star$ accordingly.*

## A.5 Limit Cases of Equation (7)

In the following, we consider limit cases of Equation (7).

### A.5.1 Value Iteration Recovered

Here, we consider $\alpha = 1$ and $\beta \to 0$. While one can easily recover value iteration as a special case by inspecting Equation (5) from the main paper simply by setting $\alpha = 1$ and $\beta = 0$, it can be insightful how to obtain Bellman's classical optimality principle as a limit case from Equation (7):

$$\lim_{\beta \to 0} V^\star(\boldsymbol{s}) =$$

$$\lim_{\beta \to 0} \beta \log \sum_{\boldsymbol{a}} \exp \left( \frac{1}{\beta} \mathcal{R}(\boldsymbol{s}, \boldsymbol{a}) + \mathbb{E}_{\mathcal{P}(\boldsymbol{s}'|\boldsymbol{s},\boldsymbol{a})} \left[ \log q^\star(\boldsymbol{a}|\boldsymbol{s}', \boldsymbol{s}) + \frac{\gamma}{\beta} V^\star(\boldsymbol{s}') \right] \right) \stackrel{\text{L'Hospital if } \max_{\boldsymbol{a}} \left( \mathcal{R}(\boldsymbol{s},\boldsymbol{a}) + \gamma \mathbb{E}_{\mathcal{P}(\boldsymbol{s}'|\boldsymbol{s},\boldsymbol{a})} \left[ V^\star(\boldsymbol{s}') \right] \right) > 0}{=}$$

$$\lim_{\beta \to 0} \frac{\sum_{\boldsymbol{a}} \exp \left( \frac{1}{\beta} \mathcal{R}(\boldsymbol{s}, \boldsymbol{a}) + \mathbb{E}_{\mathcal{P}(\boldsymbol{s}'|\boldsymbol{s},\boldsymbol{a})} \left[ \log q^\star(\boldsymbol{a}|\boldsymbol{s}', \boldsymbol{s}) + \frac{\gamma}{\beta} V^\star(\boldsymbol{s}') \right] \right) \left( \cancel{-\frac{1}{\beta^2}} \right) \left( \mathcal{R}(\boldsymbol{s}, \boldsymbol{a}) + \gamma \mathbb{E}_{\mathcal{P}(\boldsymbol{s}'|\boldsymbol{s},\boldsymbol{a})} \left[ V^\star(\boldsymbol{s}') \right] \right)}{\left( \cancel{-\frac{1}{\beta^2}} \right) \sum_{\boldsymbol{a}} \exp \left( \frac{1}{\beta} \mathcal{R}(\boldsymbol{s}, \boldsymbol{a}) + \mathbb{E}_{\mathcal{P}(\boldsymbol{s}'|\boldsymbol{s},\boldsymbol{a})} \left[ \log q^\star(\boldsymbol{a}|\boldsymbol{s}', \boldsymbol{s}) + \frac{\gamma}{\beta} V^\star(\boldsymbol{s}') \right] \right)} =$$

$$\max_{\boldsymbol{a}} \left( \mathcal{R}(\boldsymbol{s}, \boldsymbol{a}) + \gamma \mathbb{E}_{\mathcal{P}(\boldsymbol{s}'|\boldsymbol{s},\boldsymbol{a})} \left[ V^\star(\boldsymbol{s}') \right] \right).$$

The above is true if $\left( \mathcal{R}(\boldsymbol{s}, \boldsymbol{a}) + \gamma \mathbb{E}_{\mathcal{P}(\boldsymbol{s}'|\boldsymbol{s},\boldsymbol{a})} \left[ V^\star(\boldsymbol{s}') \right] \right) > 0$ for at least one action $\boldsymbol{a}$ given the state $\boldsymbol{s}$, because numerator and denominator are then dominated by the maximum sum element. If $\max_{\boldsymbol{a}} \left( \mathcal{R}(\boldsymbol{s}, \boldsymbol{a}) + \gamma \mathbb{E}_{\mathcal{P}(\boldsymbol{s}'|\boldsymbol{s},\boldsymbol{a})} \left[ V^\star(\boldsymbol{s}') \right] \right) \leq 0$ given $\boldsymbol{s}$, then one needs to focus on the second line of the above expression because L'Hospital does not apply anymore. In this case, the maximum element will dominate the sum dwarfing the non-maximum elements. As a consequence $\log$ and $\exp$ cancel each other and $\beta$ cancels with $(1/\beta)$. $\beta$ hence only multiplies with the intrinsic motivation term induced by empowerment. The latter is going to therefore vanish since $\beta \to 0$, resulting in the same expression as in the last line above.

### A.5.2 Cumulative One-Step Empowerment Recovered

Here we consider $\alpha \to 0$ and $\beta = 1$. In line with the previous section, recovering cumulative one-step empowerment can be easily obtained from Equation (5) by setting $\alpha = 0$ and $\beta = 1$. The limit case of Equation (7) is trivially given by:

$$\lim_{\alpha \to 0} V^\star(\boldsymbol{s}) =$$

$$\lim_{\alpha \to 0} \log \sum_{\boldsymbol{a}} \exp \left( \alpha \mathcal{R}(\boldsymbol{s}, \boldsymbol{a}) + \mathbb{E}_{\mathcal{P}(\boldsymbol{s}'|\boldsymbol{s},\boldsymbol{a})} \left[ \log q^\star(\boldsymbol{a}|\boldsymbol{s}', \boldsymbol{s}) + \gamma V^\star(\boldsymbol{s}') \right] \right) =$$

$$\log \sum_{\boldsymbol{a}} \exp \left( \mathbb{E}_{\mathcal{P}(\boldsymbol{s}'|\boldsymbol{s},\boldsymbol{a})} \left[ \log q^\star(\boldsymbol{a}|\boldsymbol{s}', \boldsymbol{s}) + \gamma V^\star(\boldsymbol{s}') \right] \right).$$

### A.5.3 Non-Cumulative One-Step Empowerment Recovered

In addition to $\alpha \to 0$ and $\beta = 1$ from the former section, we consider here $\gamma \to 0$ in the following:

$$\lim_{\alpha \to 0, \gamma \to 0} V^\star(\boldsymbol{s}) =$$

$$\lim_{\alpha \to 0, \gamma \to 0} \log \sum_{\boldsymbol{a}} \exp \left( \alpha \mathcal{R}(\boldsymbol{s}, \boldsymbol{a}) + \mathbb{E}_{\mathcal{P}(\boldsymbol{s}'|\boldsymbol{s},\boldsymbol{a})} \left[ \log q^\star(\boldsymbol{a}|\boldsymbol{s}', \boldsymbol{s}) + \gamma V^\star(\boldsymbol{s}') \right] \right) =$$

$$\log \sum_{\boldsymbol{a}} \exp \left( \mathbb{E}_{\mathcal{P}(\boldsymbol{s}'|\boldsymbol{s},\boldsymbol{a})} \left[ \log q^\star(\boldsymbol{a}|\boldsymbol{s}', \boldsymbol{s}) \right] \right).$$

The latter can be also obtained by running one-step empowerment ($k = 1$) according to the Blahut-Arimoto scheme from the main paper's background section in Proposition 1 until convergence, and subsequently plugging the converged solution $\pi^\star_{\text{empower}}$ from Equation (4) into Equation (2).

# B  Pseudocode for the Empowered Actor-Critic (EAC)

Let's restate the optimization objectives from Section 5.1 as functions of the optimization parameters and a batch $\mathcal{B} = \{(\boldsymbol{s}, \boldsymbol{a}, r, \boldsymbol{s}')^{(b)}\}_{b=1}^{B}$ sampled from the replay buffer, where $B$ is the batch size:

$$J_Q(\theta, \mathcal{B}) = \frac{1}{B} \sum_{b=1}^{B} \left( Q_\theta \left( \boldsymbol{s}^{(b)}, \boldsymbol{a}^{(b)} \right) - \left( \alpha r^{(b)} + \gamma V_\psi \left( \boldsymbol{s}'^{(b)} \right) \right) \right)^2,$$

$$J_V(\psi, \mathcal{B}) = \frac{1}{B} \sum_{b=1}^{B} \left( V_\psi \left( \boldsymbol{s}^{(b)} \right) - \mathbb{E}_{\pi_\phi(\boldsymbol{a}|\boldsymbol{s}^{(b)})} \left[ Q_\theta \left( \boldsymbol{s}^{(b)}, \boldsymbol{a} \right) + \beta f \left( \boldsymbol{s}^{(b)}, \boldsymbol{a} \right) \right] \right)^2,$$

$$J_\pi(\phi, \mathcal{B}) = -\frac{1}{B} \sum_{b=1}^{B} \mathbb{E}_{\pi_\phi(\boldsymbol{a}|\boldsymbol{s}^{(b)})} \left[ Q_\theta \left( \boldsymbol{s}^{(b)}, \boldsymbol{a} \right) + \beta f \left( \boldsymbol{s}^{(b)}, \boldsymbol{a} \right) \right],$$

$$J_p(\chi, \mathcal{B}) = -\frac{1}{B} \sum_{b=1}^{B} \mathbb{E}_{\pi_\phi(\boldsymbol{a}|\boldsymbol{s}^{(b)})\mathcal{P}_\xi(\boldsymbol{s}'|\boldsymbol{s}^{(b)},\boldsymbol{a})} \left[ \log p_\chi \left( \boldsymbol{a} \middle| \boldsymbol{s}', \boldsymbol{s}^{(b)} \right) \right],$$

$$J_\mathcal{P}(\xi, \mathcal{B}) = -\frac{1}{B} \sum_{b=1}^{B} \log \mathcal{P}_\xi \left( \boldsymbol{s}'^{(b)} \middle| \boldsymbol{s}^{(b)}, \boldsymbol{a}^{(b)} \right).$$

Denoting the corresponding learning rates as $\delta_\theta, \delta_\psi, \delta_\phi, \delta_\chi$ and $\delta_\xi$, we can phrase pseudocode for the empowered actor-critic conveniently.

---
**Algorithm 1** Empowered Actor-Critic (EAC)

---
initialize $\theta$, $\psi$, $\phi$, $\chi$ and $\xi$
**for** each episode **do**
    $\boldsymbol{s}_0 \leftarrow$ reset environment
    **for** each environment step $t$ **do**
        *# environment interaction*
        $\boldsymbol{a}_t \sim \pi_\phi(\boldsymbol{a}_t|\boldsymbol{s}_t)$                                   ▷ sample an action from the policy
        $r_t \leftarrow \mathcal{R}(\boldsymbol{s}_t, \boldsymbol{a}_t)$                                            ▷ evaluate the action
        $\boldsymbol{s}_{t+1} \sim \mathcal{P}(\boldsymbol{s}_{t+1}|\boldsymbol{s}_t, \boldsymbol{a}_t)$                                   ▷ execute the action
        $\mathcal{D} \leftarrow \mathcal{D} \cup \{(\boldsymbol{s}_t, \boldsymbol{a}_t, r_t, \boldsymbol{s}_{t+1})\}$           ▷ add the transition to the replay buffer
        *# gradient updates*
        $\mathcal{B} \sim \mathcal{D}$                          ▷ draw a transition batch from the replay buffer
        $\theta \leftarrow \theta - \delta_\theta \nabla_\theta J_Q(\theta, \mathcal{B})$                          ▷ update the Q-critic
        $\psi \leftarrow \psi - \delta_\psi \nabla_\psi J_V(\psi, \mathcal{B})$                        ▷ update the V-critic
        $\phi \leftarrow \phi - \delta_\phi \nabla_\phi J_\pi(\phi, \mathcal{B})$                        ▷ update the policy
        $\chi \leftarrow \chi - \delta_\chi \nabla_\chi J_p(\chi, \mathcal{B})$                    ▷ update the inverse dynamics
        $\xi \leftarrow \xi - \delta_\xi \nabla_\xi J_\mathcal{P}(\xi, \mathcal{B})$                     ▷ update the transition model
    **end for**
**end for**

---

Note that practically when updating the Q-value parameters $\theta$, we recommend replacing the value target $V_\psi$ with an exponentially averaged value target $V_{\bar{\psi}}$ instead where $\bar{\psi} \leftarrow (1 - \tau)\bar{\psi} + \tau\psi$ with horizon parameter $\tau$ —see [21].

Note also that our second proposed method, actor-critic with intrinsic empowerment (ACIE), can use the same algorithm for learning parametric function approximators by setting $\alpha = 0$ and $\beta = 1$. Since Algorithm 1 is an off-policy method that uses a replay buffer, it can be combined with any other actor-critic algorithm whose actor is collecting samples from the environment. An ACIE-agent can hence be trained concurrently and used to generate intrinsic rewards according to Equation (6) from the main paper. These intrinsic rewards are then added to the extrinsic rewards of the agent that collects samples from the environment to encourage visiting states with high cumulative empowerment.

# C Experiments

The following subsections provide a detailed description of the setups that we used for the grid world and MuJoCo experiments.

## C.1 Grid World

In the grid world setting from the main paper (Section 4.3), the agent has to reach a goal in the lower left of a $16 \times 16$ grid, which is rewarded with $+2$. The agent can execute nine actions in each grid cell: left, right, up, down, as well as diagonally or stay in place. The transition function is deterministic. The discount factor $\gamma$ was set to $\gamma = 0.95$ in the experiments. The stopping criterion for the value iteration scheme was when the infinity norm of two consecutive value vectors dropped below $\epsilon_{\text{outer}} = 5 \cdot 10^{-4}$. The stopping criterion for the inner Blahut-Arimoto scheme for each value iteration step was when the maximum absolute difference between the probability values in consecutive $q$ and $\pi_{\text{behave}}$ dropped below the threshold $\epsilon_{\text{inner}} = 5 \cdot 10^{-4}$.

Below is another grid world example similar to the one from the main paper, where the agent has to reach a goal in the upper right of a $16 \times 16$ grid. Reaching the goal is rewarded with $+1$ and terminates the episode whereas every step is penalized with $-1$. The transition function is probabilistic. Whenever the agent takes a step, the agent ends up at the intended next grid cell with only a 20%-chance. There is either a 30%-chance of a horizontal perturbation by one step, or a 30%-chance of a vertical perturbation by one step, or a 20%-chance of a diagonal perturbation by one step. The discount factor $\gamma$ was set to $\gamma = 0.6$ (leading to more myopic policies).

| Grid World | $\alpha = 0.0, \beta = 1.0$ | $\alpha = 0.2, \beta = 0.8$ | $\alpha = 0.4, \beta = 0.6$ | $\alpha = 0.6, \beta = 0.4$ | $\alpha = 0.8, \beta = 0.2$ | $\alpha = 1.0, \beta = 0.0$ |

Figure 4: Value Iteration for another Grid World Example. The figure is similar to Figure 1 from the main paper. The agent aims to arrive at the goal 'G' in the upper right. The plots show optimal values for different $\alpha$ and $\beta$ ranging from raw cumulative empowerment learning to reward maximization. Raw cumulative empowerment learning ($\alpha = 0.0$, $\beta = 1.0$, see second plot) assigns high values to states where many other states can be reached, i.e. the middle of the upper and lower room as well as the door connecting them; and low values to states where the number of reachable next states is low, i.e. close to walls and corners as well as in the bottom right dead end and the goal (because it terminates the episode). Ordinary cumulative reward maximization ($\alpha = 1.0$, $\beta = 0.0$, see rightmost plot) assigns high values to states close to the goal and low values to states that are far away.

## C.2 MuJoCo

For all our MuJoCo experiments, we followed standard literature regarding hyperparameter settings [21]. We used Adam [24] as optimizer for all parametric functions with a learning rate $\delta = 3 \cdot 10^{-4}$. The discount factor $\gamma$ was set to $\gamma = 0.99$, the replay buffer size was $5 \cdot 10^5$ and the batch size for training was 256. All neural networks were implemented in PyTorch. The critic and policy networks had two hidden layers whereas the transition and inverse dynamics model networks had three hidden layers. The number of units per hidden layer was 256 using ReLU activations. In line with [21], we used an exponentially averaged V-value target for updating Q-value parameters with a horizon parameter $\tau = 0.01$—explained at the end of Appendix B. Our specific trade-off parameters $\alpha$ and $\beta$ were set to $\alpha = 10$ and $\beta = 0.1$ respectively (both for EAC and ACIE experiments) as determined through initial experiments on InvertedDoublePendulum-v2 and HalfCheetah-v2. ACIE-generated intrinsic rewards were furthermore clipped to not exceed an absolute value of 20.

Both policy and inverse dynamics model assume that actions are distributed according to a multivariate Gaussian with diagonal covariance. They receive as input the (concatenated) vectors of $s$ and $(s, s')$ respectively. They output the mean and the log standard deviation vectors from which real-valued actions can be sampled. The real-valued actions are subsequently squashed through a sigmoid function

because MuJoCo has bounded action spaces. We used $\texttt{tanh}$ [21] scaled by the environment-specific bounds. The transition network assumes that states are distributed according to a multivariate isotropic Gaussian with a given standard deviation of $10^{-5}$. It receives as input the concatenated vectors of $(s, a)$ and outputs the mean of $s'$. The value networks merely output a single real number for cumulative reward prediction given the input. The input to the Q-value network are the concatenated vectors of $(s, a)$ whereas the input to the V-value network is $s$.

Following [67, 68, 15, 21], we used a twin Q-critic rather than a single Q-critic. This means that two Q-critic networks $Q_{\theta_1}(\cdot, \cdot')$ and $Q_{\theta_2}(\cdot, \cdot')$ are trained. When updating the V-critic and the policy, $Q_{\theta}(\cdot, \cdot')$ is replaced with $\min\{Q_{\theta_1}(\cdot, \cdot'), Q_{\theta_2}(\cdot, \cdot')\}$ to prevent value overestimation. To train the policy parameters, we applied the reparameterization trick on the actions [25, 50]—see [21] Appendix C. We also found it helpful to bound the log standard deviation of the policy and inverse dynamics networks according to [9] Appendix A.1 to make our implementation more stable.

We compare against an SAC baseline with hyperparameters chosen according to the original paper [21], except using a reward scale of 10 to ensure comparability with our methods EAC and ACIE. We furthermore compare against the DDPG and PPO baselines from RLlib [35] using hyperparameters settings following [15] and [57], but with the same neural network architectures as used in EAC, ACIE and SAC to ensure a fair comparison.

Note that in neither Figure 2 nor Figure 3 from the main paper do we report results from DDPG on Ant because the RLlib baseline implementation of that algorithm was not able to learn with our experimental protocol in that specific environment. In initial trials, we observed that DDPG in Ant leads to a rapid drop in performance to large negative values after the very first few episodes and never recovers from there within the next $5 \cdot 10^5$ environment steps. This performance pattern is in line with the experiments conducted in previous literature and can be seen by carefully inspecting Figure 1(d) from the SAC-paper [21].