[Reviews · NeurIPS 2019]

Reviewer 1



5) Originality: This work generalizes a number of previous work that combines RL and information-theoretic methods. Quality: 6) The algorithm and the theoretical results are sound. 7) In certain environments, the results reported for SAC are much lower than the ones in the original paper e.g. Half-Cheetah ~4000 vs ~8000. I would recommend investigating the source of this discrepancy. 8) Clarity: The paper is clear and the results could be replicated from the information provided. I think some of the derivations could have been made clearer, see improvements. 9) Significance: The work generalizes a number of past works and I thus think it is likely that other researchers will build on this work. Conclusion: 10) The method and the theoretical contributions are sound, but given certain anomalies in the experiments, I do not recommend acceptance at this time.

Reviewer 2



AFTER REBUTTAL ============== I thank the authors for the clarifications given in the author response, especially regarding the 1-step vs multi-step empowerment. I keep my score and recommend the paper to be accepted. Originality ----------- The paper seems to be the first to combine reward maximization and 1-step empowerment inside a single objective function. A slightly more explicit placement of the presented work in a broader context of research on empowerment in the Introduction or Motivation sections would be beneficial for the reader. Quality -------- Theoretical developments and proofs were checked selectively and did not reveal significant flaws. The basic premise of using empowerment to improve reward maximization may be somewhat questionable. Especially given that empowerment is expensive to compute, perhaps a better argument would be to consider a multi-task setting, where task transfer can benefit from the agent being initially empowered. Adding a paragraph detailing drawbacks of the proposed approach would be beneficial. For example, learning forward and inverse models may introduce bias and hinder performance of model-free methods. Was such effect observed in the experiments? Clarity ------- The paper is written clearly and structured well. Significance -------------- The main contribution of the paper is on the theoretical side. An important question which was not addressed by the paper is how much is lost by only considering 1-step empowerment. Since experiments were carried out in a single-task RL setting, the benefit of using the empowerment were not so clear (see Fig. 2). In general, maybe the whole line or argumentation in the paper can be slightly adapted to better motivate the combination of the reward with empowerment. It seems more plausible to expect gains in a multi-task setting.

Reviewer 3



This is a simple paper with a straightforward proposal, execution, and presentation of results. The idea of combining rewards and empowerment has been up for grabs for some time, and thus it is by itself not particularly surprising nor controversial. (Ironically, the idea is actually conceptually at odds with the original, explicit proposal to do away altogether with rewards and to replace it with a universal objective—that's why empowerment got invented in the first place.) I don't have many remarks, just some minor details. - Section 2.2. (Empowerment'') could be simpler (mainly in notation). Equation (2) is slightly confusing at first, as \pi_empower appears explicitly in the denominator inside the log, but only implicitly in the numerator. The lemmas that follow are trivial. - Similarly, section 4.1. (Existence of Unique...'') could also be simpler. In fact, I was surprised that this hadn't been shown before. It's nice to see it spelled out. - Section 4.3 with the grid-world example. This example is not very illuminating. - Section 5.2 (Experiments with Deep Function Approximators''): The experimental results did not look very convincing to me. Empowerment either seems to add little or even deteriorates the policies found by SAC. Perhaps the MuJoCo domains were not great for showcasing the method. In general, while I did enjoy the paper, I felt that the significance of the results were modest, firstly because the ideas are not surprising, and secondly because I did not feel that they made much of a difference to the state of the art. PS: The bibliography is surprisingly complete. Were all the references woven into the text? I felt they weren't. *** POST REBUTTAL COMMENTS *** I would like to thank the authors for addressing all the comments. I went through the experimental results again, and decided to increase the score.

[Author Response · NeurIPS 2019]

**R1.** Thank you for appreciating the theoretical contribution and significance. Your main concern seems the performance difference in our soft actor-critic (SAC) compared to the SAC from [20,21] on HalfCheetah. This was earlier noted by other researchers who tried to reproduce SAC results, leading to a GitHub issue which has been resolved just recently [Q. Vuong. Unable to reproduce result on HalfCheetah-v2. In *GitHub rail-berkeley/softlearning Repository*, Issue #75, 2019.]. One of the SAC authors of [21] was able to reproduce low SAC performance on HalfCheetah for a specific seed setting *confirming that the effect is of statistical nature only and scientifically valid*. Details follow next. First note that the original SAC paper [20] used Mujoco v1-environments, while we used the latest v2-versions. Since then, SAC has been evaluated on v2 by the same authors as in [20]—see Figure 1 in [21]. While the performance of our SAC is comparable with the SAC from [21] on Hopper-v2, Walker2d-v2, Ant-v2 and Humanoid-v2 after 500k steps, there is a discrepancy on HalfCheetah-v2. Others obtained similar reproducing results leading to the aforementioned GitHub issue. The first figure in the first comment of the GitHub issue reports similar HalfCheetah results as we do. One SAC-paper author [21] was able to reproduce lower-performance results (see comment from May 19). The final conclusion was that this is caused by the cheetah occasionally flipping over (see comment from May 20) for specific seed settings apparently different from those used in [21]. The comment from June 24 resolves the issue: proper randomization of both environment creation and action sampling seeds in OpenAI gym leads to low performance, whereas clamping the action sampling seed to 0 yields high performance. On a minor note, the SAC results in [20,21] were obtained by averaging over 5 seeds which is not enough in Mujoco—different pairs of 5 seeds can yield significantly different results, see [43] slide 21. Therefore, we conducted experiments with 10 seeds. Hence, our evaluation is statistically more sound compared to the SAC papers [20,21]. We hope this clarification addresses your main concern.

**Improvements. 11)** Eq. (7) pre-states our main theoretical result in advance, but the rest of Section 4 is required to understand it—we clarified that. In short, under optimal $V^\star$ and $q^\star$, the optimal policy $\pi^\star$ for the second line in Eq. (7) is given by Eq. (9). Plugging $\pi^\star$ back into the second line of Eq. (7) (but without the max-operator and assuming an optimal $q^\star$) yields the third line. If $q^\star$ was replaced with a fixed $q$ that does not depend on the next state $s'$, then cumulative KL regularization is recovered in which case the third line of Eq. (7) is a lower bound to the ordinary MDP formulation without logarithmic penalty. But the lower-bound statement does not hold for empowerment regularization because it *adds* intrinsic reward while KL regularization *subtracts* intrinsic reward on average. **12)** You are right. Eq. (9) can be re-formulated to yield a result similar to what you mentioned for empowerment regularization. **13)** Figure 2 adopts the performance metric from [8]. We also report the plot you suggested with mean episodic reward curves in Appendix Figure 3 for all environments (we can swap them). We adjusted the paper w.r.t. **11) - 13)** accordingly.

**R2.** Thank you for valuing our theoretical contribution. Your main concern seems that we use a 1-step rather than a multi-step empowerment formulation. We need to stress that we optimize for *cumulative* and not *instantaneous* 1-step empowerment. Cumulative 1-step empowerment yields *non-myopic* agents and *has similar properties as multi-step empowerment*. Note that in Figure 1 in the paper, $\gamma$ was 0.6 which might evoke the impression of a myopic policy in the second plot. Below is a more illuminating example with $\gamma = 0.95$ (also with $\alpha = 0$ and $\beta = 1$).

Not requiring a multi-step policy is actually a strength because a multi-step policy executes a sequence of actions (and cannot "correct" for an action when observing another state in the meanwhile). We hope this addresses your main concern and we would be happy if you shared your enthusiasm with the other reviewers (we added the new example to the paper).

**Improvements. Multi-Task)** We agree, empowerment could be particularly beneficial for multi-task (but this is outside the scope of the rebuttal). **Drawbacks)** We have not investigated model biases. However, empowerment specifies a particular optimization objective, and one can design reward signals that conflict with empowerment signals (e.g. negative empowerment). This could explain hindered performance—we clarified that. **Code)** We triggered the internal process for code release (in a non-academic institution, there are intellectual property regulations). **Run Time / Complexity)** Theorem 2 says for how many iterations $i$ the value iteration needs to run to guarantee optimal values with epsilon-precision ($i \geq \log_\gamma(\epsilon(1-\gamma)/const)$). Proposition 3 says that one value iteration step (for a particular state $s$) requires an iterative Blahut-Arimoto scheme that converges at a rate of $O(1/j)$ where $j$ is the number of "inner" iterations. Similarly to the "outer" value iteration scheme, it can be determined how many inner iterations $j$ are required to obtain epsilon-precision, i.e. with the proof of Appendix Lemma 5: $j \geq (\beta/\epsilon)const$. The complexity of the inner Blahut-Arimoto scheme for a single state $s$ is $O(j|S||A|)$—see Eqs. (13,14). The overall complexity is $O(ij|S|^2|A|)$. We adjusted the paper accordingly.

**R3.** Thank you for your feedback. You seem concerned about the Mujoco results. While we consider the generalized MDP formulation plus theory as our main contribution, the experiments show for the first time that empowerment can improve RL in high-dimensional tasks. As you mentioned, this has been a long-standing research question not addressed before. *Empowerment leads to significant improvements in 6 tasks (most notably Ant which is amongst the most difficult tasks), and in the other 2 still to initial improvements* compared to the state-of-the-art SAC—see Figure 2. While we agree that more environments are always better, we already provide a suite of 8 different environments where others usually report less, e.g. 6 in SAC [20]. We adjusted the paper to address your minor details.

[Meta-Review · NeurIPS 2019]

We all liked the submission and recommend it is accepted.